# Latest Altimetry-Based Sea Ice Freeboard and Volume Inter-Annual Variability in the Antarctic over 2003–2020

**Florent Garnier** [1,*,†] [ID], **Marion Bocquet** [1,†] [ID], **Sara Fleury** [1,†] [ID], **Jérôme Bouffard** [2,†] [ID], **Michel Tsamados** [3,†] [ID], **Frédérique Remy** [1,†], **Gilles Garric** [4,†] and **Aliette Chenal** [4,†]

1   Laboratoire d'Etudes en Géophysique et Océanographie Spatiales (LEGOS), CNRS/UMR5566, Université Paul Sabbatier, 31400 Toulouse, France
2   European Space Agency (ESA), Earth Observation Directorate, Via Galileo Galilei, 2-00044 Frascati, Italy
3   Centre for Polar Observation and Modelling, Department of Earth Sciences, University College London, London WC1E 6BT, UK
4   Mercator Ocean, 31520 Ramonville Saint Agne, France
*   Correspondence: florent.garnier@legos.obs-mip.fr
†   These authors contributed equally to this work.

**Abstract:** The relatively stable conditions of the sea ice cover in the Antarctic, observed for almost 40 years, seem to be changing recently. Therefore, it is essential to provide sea ice thickness (SIT) and volume (SIV) estimates in order to anticipate potential multi-scale changes in the Antarctic sea ice. For that purpose, the main objectives of this work are: (1) to assess a new sea ice freeboard, thickness and volume altimetry dataset over 2003–2020 and (2) to identify first order impacts of the sea ice recent conditions. To produce these series, we use a neuronal network to calibrate Envisat radar freeboards onto CryoSat-2 (CS2). This method addresses the impacts of surface roughness on Low Resolution Mode (LRM) measurements. During the 2011 common flight period, we found a mean deviation between Envisat and CryoSat-2 radar freeboards by about 0.5 cm. Using the Advanced Microwave Scanning Radiometer (AMSR) and the dual-frequency Altimetric Snow Depth (ASD) data, our solutions are compared with the Upward looking sonar (ULS) draft data, some in-situ measurement of the SIMBA campaign, the total freeboards of 6 Operation Ice Bridge (OIB) missions and ICESat-2 total freeboards. Over 2003–2020, the global mean radar freeboard decreased by about −14% per decade and the SIT and SIV by about −10% per decade (considering a snow depth climatology). This is marked by a slight increase through 2015, which is directly followed by a strong decrease in 2016. Thereafter, freeboards generally remained low and even continued to decrease in some regions such as the Weddell sea. Considering the 2013–2020 period, for which the ASD data are available, radar freeboards and SIT decreased by about −40% per decade. The SIV decreased by about −60% per decade. After 2016, the low SIT values contrast with the sea ice extent that has rather increased again, reaching near-average values in winter 2020. The regional analysis underlines that such thinning (from 2016) occurs in all regions except the Amundsen-Bellingshausen sea sector. Meanwhile, we observed a reversal of the main regional trends from 2016, which may be the signature of significant ongoing changes in the Antarctic sea ice.

**Keywords:** sea ice thickness; sea ice volume; CryoSat-2; envisat; climate change

## 1. Introduction

Observations of sea ice thickness (SIT) and concentration (SIC) in the Arctic have been extensively analysed in the last decades e.g., [1–4]. They have demonstrated the accelerating sea ice decline e.g., [5,6], its link with the global warming e.g., [7,8], and have emphasised the impacts on the whole climate system e.g., [9–12]. In the Antarctic, the situation contrasts with that in the Arctic. The ≈40 years of sea ice extent from passive microwave satellite measurements have recorded a global decades-long slight increase, to reach a record high in 2014 [13,14]. This gradual positive trend, by ≈1.5% per decade, have large

regional differences and variabilities. In particular in the Amundsen-Bellingshausen (A-B) sector, where the sea ice has reduced [15]. This decrease, associated with an increase in the Ross Sea, is very likely to be linked with a deepening of the Amundsen Sea Low (ASL) pressure system that drives warm northerly flows in the A-B area and cold winds in the Ross Sea e.g., [16,17]. Although anomalies of meridional winds are essential to explain sea ice interannual to multi-decadal variabilities e.g., [18,19], the global increase seems to be mainly driven by the negative phase of the Interdecadal Pacific Oscillation [20]. Explanations based on enhanced basal melting of ice shelves [21] or the stratospheric ozone depletion [22] have been denied by Swart and Fyfe [23] and Bitz and Polvani [24].

The 2014 maximum high was directly followed by a very rapid decline, to reach the lowest sea ice coverage of the observations in 2016–2017 [15]. The factors that contributed to this abrupt decrease, equivalent to 30 years of sea-ice loss in the Arctic [25], are multiple and still being questioned. Stuecker et al. [26] state that it is linked with SST anomalies induced by the strong 2015/2016 El Niño event and the particularly low SAM index. Schlosser et al. [27] add that an anomalous strong meridional atmospheric flow has caused southward heat advection. This anomalous circulation is probably partly driven by high tropical convections in the Indian and the Pacific oceans due to anthropogenic forcing [28]. In addition, the positive convective heating anomalies in the eastern Indian/western Pacific ocean, associated with record Sea Surface Temperatures (SSTs) and precipitations in late summer 2016, would have produced an anomalous Rossby wave response. This could have led to the transport of warm waters southwards [29]. The potential role of thermodynamic surface forcings was also analysed in Kusahara et al. [30].

Since this minimum, the winter sea ice extent has increased again. In 2021, it was consistent with the average coverage of the last 4 decades of satellite observation.

Overall, observations of sea ice thickness in the Antarctic remain space and time limited, and correlations with sea ice extent variations have not yet been analysed. Besides specific field campaigns, such as the Sea Ice Mass Balance in the Antarctic (SIMBA) [31] or the Antarctic Remote Ice Sensing Experiment (ARISE) [32,33], the main source of informations was long provided by local measurements from Upward looking Sonar (ULS) [34,35], in-situ drilling profiles [36] or ship-based electromagnetic-inductive (EM) sounding [37–39]. The Antarctic Sea ice Processes and Climate (ASPEcT) program set up from the Scientific Committee on Antarctic Research (SCAR) has archived about 2 decades of data from the logs of icebreakers [40]. However, as shipping routes preferably go through thinner sea ice, these measurements are generally biased low [41]. From 2009, several NASA's Operations IceBridge (OIB) campaigns have routinely measured surface elevations from airborne remote sensing e.g., [42,43]. In winter 2017/2018, an ESA CRYOsat Validation EXperiment (CryoVex) airborne campaign, including Ka-band and Ku-band radar measurements, was also conducted in the Antarctic [44].

At global scale, first sea ice elevations was made from the European Remote-Sensing (ERS) satellite measurements [41]. They demonstrated the potential for radar altimetry to provide relevant informations on sea ice thickness in the Antarctic. Later, Price et al. [45] evaluated first CryoSat-2 (CS2) freeboard estimates in the McMurdo sound by comparisons with in-situ data of the New Zealand field campaigns in 2011 and 2013. Overall, they found a positive bias by about 14 cm, indicating a radar echo reflection in the snow layer rather than at the snow/ice interface over fast ice. Within the European Space Agency (ESA) Climate Change Initiative (CCI), Schwegmann et al. [46] and Paul et al. [47] computed Envisat and CS2 sea ice freeboard and thickness over the entire Antarctic for the first time (see Sections 2 and 3.1). This dataset, which ends in 2016, has not been subsequently validated or analysed.

Meanwhile, Zwally et al. [48] first demonstrate the ability of the Ice, Cloud, and Land Elevation Satellite (ICESat) laser altimetry measurements to produce sea ice freeboards at basin scale. They estimated the sea ice thickness in the Weddell sea by combining with the passive microwave-derived snow depths estimations of the Advanced Microwave Scanning Radiometer-Earth Observing System (AMSR-E) satellite. This methodology was then

applied to all ICESat measurements. They analysed the seasonal and inter-annual variabilities of the Weddell sea [49], and validated the freeboards by comparison with the ARISE in-situ data in East Antarctica [50]. Assuming a zero ice freeboard, Kurtz and Markus [51] proposed ICESat-only SIT spatial distribution maps of the entire Antarctic and pointed out possible slow sea ice thinning in all regions, except A-B. This simple approach, which implies permanent flooding everywhere, has nevertheless rapidly shown its limitation e.g., [43,52]. Instead of using the zero ice freeboard assumption, Kwok and Kacimi [53] demonstrate the potential of combining radar and laser altimetry. Their comparisons of CS2 radar freeboards with OIB transects in the Weddell Sea indicate a mean overestimation. It again suggests a radar echo reflection above the snow–ice interface. Following this work, Kacimi and Kwok [54] examined the large scale seasonal cycle of Antarctic sea ice freeboard using CS2 and ICESat-2 (IS2) and Xu et al. [55] provide first elements for a potential thinning of the Antarctic sea ice.

To date, the main efforts to retrieve SIT in the Antarctic have relied on laser altimetry and attempts to use radar altimetry (such as those initiated by the CCI ) have not been continued. This is partly because the interpretations of radar-snow interactions in the Antarctic is further complicated by a thicker snow cover, thinner ice with more moist, and basal saline layers, or different snow microstructure and density. The lack of repeated in-situ missions, as well as the apparent lack of response to climate modifications (until 2016) also certainly explains the greater attraction to the Arctic. Since 2016–2017, things are changing, revealing near-future potential modifications of the Antarctic sea ice system. The sea ice community has since started to push for more works to improve our understanding of climate projections in the Antarctic. In that respect, the current ESA CryoSat+ Antarctic Ocean (CSAO+) project (http://cryosat.mssl.ucl.ac.uk/csao/index.html, (accessed on 1 May 2022)) aims at producing a reference CS2 sea ice freeboard and thickness product, validated against a comprehensive dataset of airborne and in-situ measurements. The present study is exactly conducted within this context. The main objectives are: (1) to present and assess a long-term sea ice freeboard, thickness and volume from altimetry over the Envisat/CryoSat-2 2003–2020 time period and (2) to provide trends and identify first order impacts of the recent conditions.

For that purpose, we first present CryoSat-2 radar freeboards computed from the Threshold First Maximum Retracker Algorithm (TFMRA) and the SAMOSA+ retrackers (cf. Appendix B) and compare with the CCI dataset. Due to the Low Resolution Mode (LRM) of the Envisat altimeter, a calibration on the CryoSat-2 solution is necessary to derive consistent freeboard from Envisat measurements [56]. The second part shows the calibration method and the resulting Envisat radar freeboards. In the third part, we compare these estimations with in-situ transects of the SIMBA campaign, ULS sea ice drafts measurements, 6 OIB airborne missions and ICESat-2 laser freeboards. The next section proposes a global and regional assessment of the freeboard, thickness and volume annual mean trends over 2003–2020. Finally, some of the main results of this study are discussed in the last part of the paper.

## 2. Materials and Methods

### 2.1. Altimetry Data

#### 2.1.1. CryoSat-2

The CryoSat-2 (CS2) data used in this study are developed at the LEGOS within the framework of the ESA CSAO+ project from the methodology described in Appendix B. Data are provided onto $500 \times 500$ EASE2 grids with a 12.5 km pixel size resolution on a monthly basis for the six month of winter (from May to October). The along-track freeboards calculated with the TFMRA50 (CS2-TFMRA) use the waveforms of the ESA L1b Baseline-D Synthetic Aperture Radar (SAR) and SARIn modes product [57]. The SAMOSA+ physical retracker (CS2-S+) freeboard solutions are derived from the heights computed at the ESA Grid Processing On Demand computing Sciences (GPOD) SARvatore chain [58].



Consistently with the study of Laforge et al. [59], the zero-padding and Hamming window are the only processing options activated for SAMOSA+.

### 2.1.2. Envisat

The Envisat radar freeboards are computed at the LEGOS laboratory using the methodology developed in the framework of the ESA Fundamental Data Records for Altimetry (FRD4ALT) project (https://www.fdr4alt.org/, (accessed on 15 February 2022)) to produce freeboards from the European-Remote Sensing Satellite (ERS) measurements in the Arctic. We use the Low Resolution Mode (LRM) waveforms of the Envisat RA-2 Sensor Geophysical Data Record (SGDR), Level 1b (L1b), version 3 product from ESA (https://doi.org/10.5270/EN1-85m0a7b, (accessed on 1 January 2022)). To account for the difference of footprint with CS2 e.g., [47,56], Envisat freeboards are calibrated on CS2 using a neural network approach (cf. Section 3.2). Freeboard and thickness data are provided over 2003–2011 (for the six winter months) on a similar monthly EASE2 grid format as CS2.

### 2.2. Snow Depth Data
### 2.2.1. Alti Snow Depth

The dual-frequency Ka-Ku Alti Snow Depth product (ASD) is fully described in Garnier et al. [60]. It was developed at the LEGOS laboratory in the framework of the ESA's CSAO+ and Polar+ Snow on Ice projects. It is freely available on the AVISO+/ODATIS national data centres (http://ctoh.legos.obs-mip.fr/data/sea-ice-products, (accessed in 1 March 2022)). The calculation rely on the difference of penetration between the Ka-band altimeter of SARAL and the Ku-band altimeter of CS2. We assume that Ku-band radar echoes fully penetrate the snow/ice interface and that the difference between these two surface elevations is only due to the penetration. Snow depth data are only provided in monthly means onto an identical EASE2 grids over 2013–2020 (only in winter). Since Garnier et al. [60], the CS2 pseudo-LRM (pLRM) waveforms are now issued from the L1b Baseline-C extracted from ESA Geophysical Ocean Product (GOP, https://earth.esa.int/documents/10174/125272/CryoSat-Baseline-C-Ocean-Product-Handbook, (accessed in 1 March 2022)) for the whole time period. It allows to have snow depth estimations in SARin mode zones.

### 2.2.2. Advanced Microwave Scanning Radiometer Data

We use the Advanced Microwave Scanning Radiometer (AMSR-E/AMSR-2) unified L3 daily snow depth data in 12.5 km × 12.5 km stereopolar grids data available at the NSIDC [https://nsidc.org/data/AU_SI12/versions/1, (accessed in 30 June 2022)] [61]. In addition, we use a snow depth climatology developed at the University of Bremen (UB) and the Institute of Environmental Physics (IUP) in the context of the CCI from daily AMSR-E and AMSR-2 snow depth data.

### 2.3. Auxillary Data
### 2.3.1. ICESat-2 Data

Since September 2018, The NASA's Ice, Cloud, and land Elevation Satellite-2 (ICESat-2) provide high resolution laser altimetry surface elevation measurements from the Advanced Topographic Laser Altimeter System (ATLAS) photon-counting lidar altimeter. e.g., [62–64] In this study, we use the ATL10/ICESat-2 L3A sea ice freeboard dataset https://earthdata.nasa.gov/ described in [65] (accessed in 30 June 2022).

### 2.3.2. Upward Looking Sonar Data

The Upward looking Sonar (ULS) dataset [35], operated by AWI, provides mooring measurements of sea ice draft in the Weddell Sea. The principle is to measure the returning time of sound pulses towards the sea surface reflected by the underside of the sea ice or the water/air interface. Data are collected at 13 fixed positions from November 1990 to March 2008 and from one mooring between March 2008 and January 2011 [66]. They are freely

available at https://doi.pangaea.de/10.1594/PANGAEA.785565 (accessed in 30 June 2022). Two methods are used to consider the sound velocity variations. The first method uses a correction line interpolated between leads (referring to the sea level) while the second method uses a sound speed model propagation. Only data corrected from the correction line are presented in this study since it is shown to be more accurate [35].

### 2.3.3. Operation Ice Bridge Data

The data of the Operation Ice Bridge (OIB) missions analysed here are provided by the NSIDC [https://nsidc.org/data/IDCSI4/versions/1] (accessed in 30 June 2022) [67]. They are also part of the data considered in the framework of the ESA CSAO+ project. It contains total freeboard estimates of the 2009 to 2013 OIB missions derived from elevations of the Airborne Topographic Mapper (ATM) lidar with the approach of Kwok et al. [68]. Unfortunately, most of the OIB campaigns in the Antarctic took place in Summer (March and April). Only 3 missions in 2009 and 3 others in 2010 took place in October, at a period when altimetry observations are available. The freeboards of the other missions have not been yet processed for a public access, and only surface elevations are available Kwok and Kacimi [53].

### 2.3.4. The Sea Ice Mass Balance in the Antarctic Campaign

The Sea Ice Mass Balance in the Antarctic (SIMBA) expedition was carried out from September to October 2007 in the Bellingshausen Sea to set-up a baseline data of sea ice and snow properties during the winter/spring transition [31]. The dataset provides in-situ observations of sea ice and snow properties at 3 short-term stations and at a 27 days drifting Ice Station Belgica (ISB) from snow pits, transect lines, ice core ans ice mass-balance buoys. It is available at https://doi.pangaea.de/10.1594/PANGAEA.815856 (accessed in 30 June 2022).

### 2.3.5. Sea Ice-Climate Change Initiative Dataset

This dataset was developed within the ESA Sea Ice-Climate Change Initiative (SI-CCI) project (updated in the CCI2). It is freely available on the CEDA Archive https://archive.ceda.ac.uk/ (accessed in 30 June 2022). The dataset includes radar freeboard and sea ice thickness estimations in the Antarctic over 2003–2016 [46,47].

### 2.3.6. Sea Ice Concentration

The monthly sea ice extent (SIE) and concentration (SIC) data are provided from the National Snow and Ice Data Center (NSIDC-0051) dataset of the sea ice index archive [https://nsidc.org/data/g02135] [69], freely available at ftp://sidads.colorado.edu/DATASETS/NOAA/G02135 (accessed in 30 June 2022). They are derived from passive microwave brightness temperature signatures from the Special Sensor Microwave Imager (SSM/I, until 2007) and the Special Sensor Microwave Imager Sounder (SSMIS, from 2008 to present) [70] of the Defense Meteorological Satellite Program (DMSP) satellites.

## 3. Results

### 3.1. Cryosat-2 Radar Freeboard

In this section, we present and analyse CryoSat-2 radar freeboards estimated from the methodology presented in Appendix B. Figure 1 presents the climatological mean and variability (standard deviation, Sd) maps of CryoSat-2 radar freeboards of the TFMRA-50 (CS2-TFMRA50) and the SAMOSA+ (CS2-S+) retracker solutions computed using all the monthly mean radar freeboard maps (from May to October) between 2011 and 2020. They are also compared with the SI-CCI dataset (Third). In addition, the maps presenting the evolution of the radar freeboard spatial distributions in winter 2011 are provided in Appendix A, Figure A1.

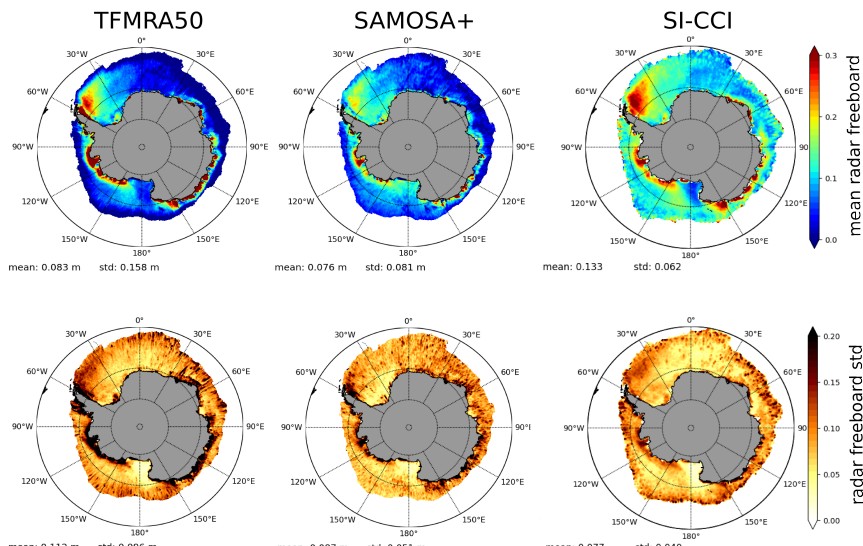

**Figure 1.** Maps of the climatological mean and standard deviation (Sd) of CryoSat-2 radar freeboards computed from the TFMRA-50 (CS2-TFMRA50, first column) and SAMOSA+ (CS2-S+, second column) retrackers over 2011–2020. They are compared with the climatological mean and standard deviation of the SI-CCI CryoSat-2 radar freeboard dataset (third column), computed over 2011–2016. Note that the climatological mean and standard deviation are calculated at each grid point using all the available monthly radar freeboard mean maps over 2011–2020.

We found coherent spatial distributions between CS2-TFMRA50 and CS2-S+ with a climatological mean difference by about 1.5 cm. Note that it is calculated (such as in the rest of the manuscript) from the mean of the radar freeboard differences at each grid point. It contrast with the difference of climatological mean, that would be the difference of the spatial mean of the maps (here 8.3 cm − 7.6 cm = 0.7 cm). The spatial distributions are characterized by thick radar freeboards in the West Weddell zone, the Bell/Amundsen sea, and along the coastal Pacific and Indian ocean areas (see Figure 8, for the geographical location of the regions). Radar freeboards are significantly smaller everywhere else. They also have equivalent patterns of variability, with magnitudes consistent with the spatial repartition. However, CS2-S+ thick radar freeboard patterns are thinner (such as in the Weddell zone) while thin patterns are thicker. This result is in agreement with the study of Laforge et al. [59] in the Arctic.

Compared to the SI-CCI product, CS2-TFMRA50 and CS2-S+ radar freeboards are globally biased low. A mean difference by about 5-6 cm is found but the spatial distributions are still consistent. One major difference with the CS2-TFMRA50 data is that the SI-CCI radar freeboards are computed using a 40% threshold, while we use 50%. Ricker et al. [71] showed that it implies a mean positive bias by about 4 cm over First Year Ice (FYI) and 6 cm over Multi-Year Ice (MYI) in the Arctic, which is coherent with the differences found here. The SI-CCI product has also a coarser grid resolution, which could explain the lower values of thick ice in narrow coastal patterns surrounded by strong gradients (such as in the Pacific and the A-B sectors). Note that the limitation to 2016 probably enhances this bias high (cf. Section 3.4). This could also partly explain the lower variabilities.

In the two estimations from TFMRA, it is relevant to observe permanent very thin radar freeboards seawards the Ross continental ice shelf. This is the signature of thin new ice exported from the large polynyas of the Ross sea, that are induced by strong continental winds flowing seawards e.g., [54,72]. On the contrary, CS2-S+ estimations do not reproduce this feature. Equivalently, the small patterns of thin new ice in front of the Ronne and the Brunt ice shelves and along the southern coast of the Larsen ice shelf e.g., [73] are less

pronounced in the CS2-S+ data. This feature might be partially due to roughness biases, as demonstrated by Landy et al. [74].

To analyse the freeboard series over the CryoSat-2 time period, Figure 2 presents maps of radar freeboard trends (slope of the linear interpolation at each grid point) computed over 2011–2020 for CS2-S+ and CS2-TFMRA50 and over 2011–2015 for the SI-CCI product. In addition, CS2-TFMRA50 trends are also computed over 2011–2015 and 2016–2020.

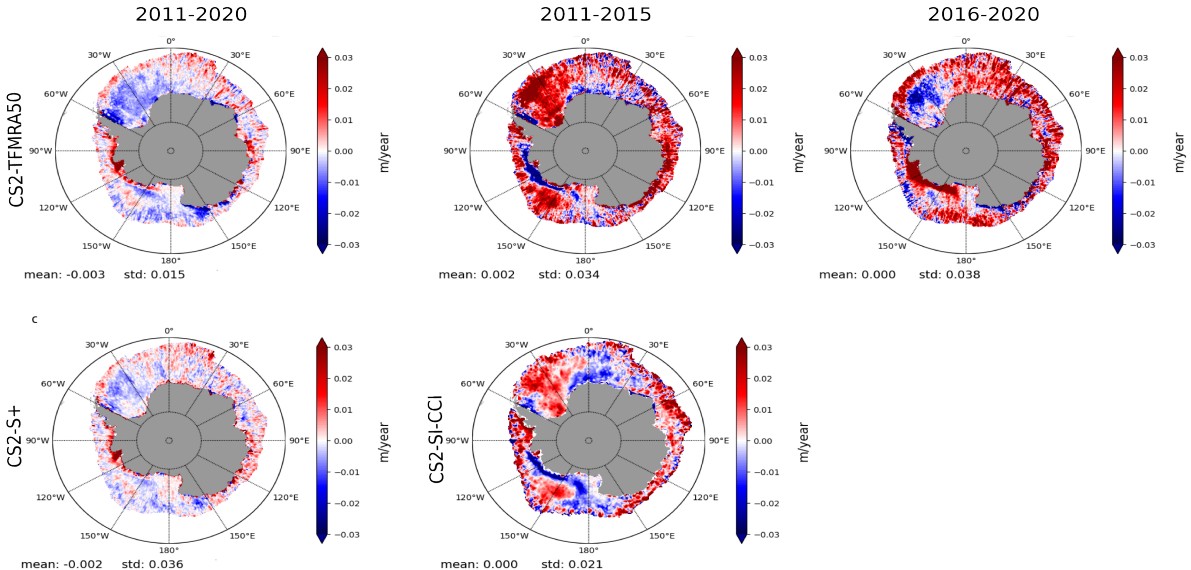

**Figure 2.** Maps of radar freeboards trends for the 2011–2020 (first column), the 2011–2015 (second column) and the 2016–2020 (third column) time periods. The 3 maps of the first line are derived from the CryoSat-2 TFMRA-50 (CS2-TFMRA50) dataset. The second line presents the maps obtained from CryoSat-2 SAMOSA+ (CS2-S+, first column)) and the SI-CCI radar freeboard dataset (second column).

Over 2011–2020 time period, the radar freeboard mean trends of CS2-TFMRA50 and CS2-S+ are similar, by about −2, −3 cm (≈30%) per decade. The two trend maps have equivalent spatial distributions, with associated strong regional differences. We observe negative trends in the Weddell sea and positive trends in the coastal A-B zone. Significant patterns of negative trends are also found along the Oate land coasts and the westward Cape Adare, in the Northwestern corner of the Ross Sea. It might indicate that the sea ice drifted from the Ross sea [54] tends to be thinner.

Over 2011–2015, the SI-CCI and CS2-TFMRA50 maps are very similar, with a mean slight increase by about 2 cm per decade. Compared with the 2011–2020 trend maps, it is striking to observe a reversal in regional behaviours almost everywhere. This feature highlights the anomalous 2016 sea ice retreat e.g., [75], as well as its impact thereafter. The 2016–2020 spatial distributions of the CS2-TFMRA50 trend maps seem to indicate that this inversion was initiated in 2016 and has lasted through 2020. In fact, these results suggest that the decrease of sea ice extent in 2016 was associated with a global mean thinning and a reversal of most regional trends, that could have impacts on a longer time scale. Further investigations on trends are provided in Section 3.4.

### 3.2. Envisat Radar Freeboard

#### 3.2.1. Re-Calibration on CryoSat-2

Compared to the ≈300 m along-track footprint of the SIRAL altimeter of CS2, the pulse limited altimeter of Envisat has a resolution comprised between 2 and 10 km. As a consequence, the impacts of surface roughness and off-nadir reflections on CS2 measurements are considerably reduced. This mainly explains that Envisat freeboards, as computed from usual heuristic approaches, do not provide consistent solutions (cf. Figure 3). Physical re-tracking methods can deal with LRM waveforms specificities, but they are not yet available

over sea ice. An alternative solution is to calibrate Envisat radar freeboard estimations on CS2 using the common flight period. In this context, Guerreiro et al. [56] proposed a calibration based only on the relation between the pulse peakiness of Envisat (*PP*) and the radar freeboard differences between Envisat and CryoSat-2. In order to take into account multiple parameters (not only the *PP*), we use a neuronal approach developed by [76] in the context of the FDR4ALT project.

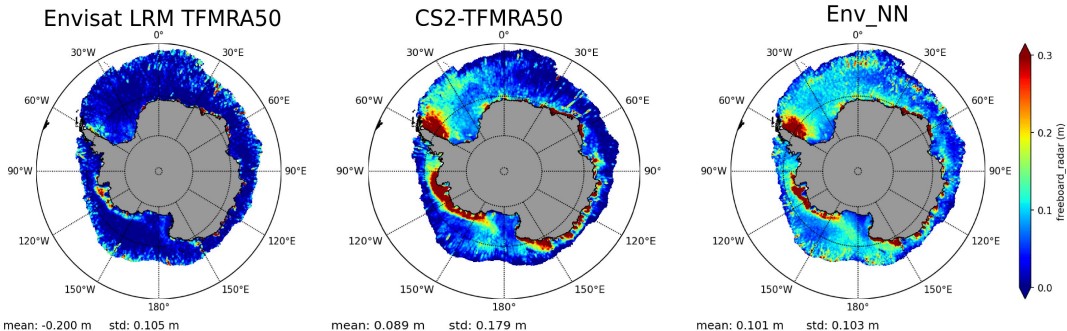

**Figure 3.** Maps of annual mean radar freeboard of Envisat LRM TFMRA50 (**left**), CS2-TFMRA50 (**middle**) and the Envisat LRM data calibrated on CS2-TFMRA50 using the neuronal network approach (Env-NN, **right**), for the 2011 common flight period.

The correction model is a multi-layer perceptron neural network that has the specification to handle well with non-linearities. By considering the data of the common winter 2011 (the training period), it computes a calibration function from the TFMRA-50 radar freeboards of CS2 and various Envisat parameters. The calibration function characterizes, at each grid point, the state of the sea ice from the Envisat inputs: LRM radar freeboard, date (month), sea ice concentration (SIC), leading edge slope, pulse peakiness and sea ice type. The correction model is then able to retrieve the radar freeboards of any month of the Envisat flight period. Note that our analysis suggests that such calibration techniques are only relevant when the freeboards are computed with the same methodology. In particular, this means with an identical retracking method. Indeed, the response to sea ice properties such as roughness can be very different from one retracker to another. Thus, Envisat radar freeboards are not calibrated on SAMOSA+ Cryosat-2 data.

Figure 3 and Table 1 highlight the impacts of the multi-layer perceptron neural network calibration on Envisat radar freeboards. The calibration method allows to reproduce the relevant radar freeboard patterns of CryoSat-2, identified in Section 3.1. The mean deviation, between Envisat and CS2, of the annual mean radar freeboard is only by about 0.5 cm, with associated identical mean spatial distributions.

**Table 1.** Summary table of monthly sea ice radar freeboard statistics of Cryosat-2 and the calibrated version of Envisat during the May–October 2011 period. The mean, the standard deviation (Sd), the RMSD and the Pearson correlation coefficient R are calculated for sea ice concentration > 75%.

|  | Mean (cm) | | Sd (cm) | | RMSD (cm) | R |
|---|---|---|---|---|---|---|
|  | CS2 | ENV_NN | CS2 | ENV_NN | CS2 vs. ENV_NN | CS2 vs. ENV_NN |
| May | 9.9 | 10 | 20.7 | 12.8 | 14.9 | 0.62 |
| June | 9.7 | 9.9 | 21.4 | 12.2 | 14.4 | 0.64 |
| July | 9.7 | 10.2 | 19.5 | 12.2 | 13.0 | 0.63 |
| August | 10.6 | 11.3 | 21.1 | 12.9 | 13.7 | 0.63 |
| September | 10.7 | 10.9 | 20.7 | 13.4 | 13.7 | 0.69 |
| October | 9.6 | 11 | 21.7 | 14 | 14.8 | 0.67 |
| Mean | 10.0 | 10.5 | 20.8 | 12.9 | 14.1 | 0.65 |

This seems slightly better than the 2.7 cm mean deviations of Paul et al. [47] conducted within the SI-CCI. Similarly with Schwegmann et al. [46] and Paul et al. [47], the calibrated Envisat estimates have higher thin ice and thinner thick ice compared to CS2 (see also Figure 4).

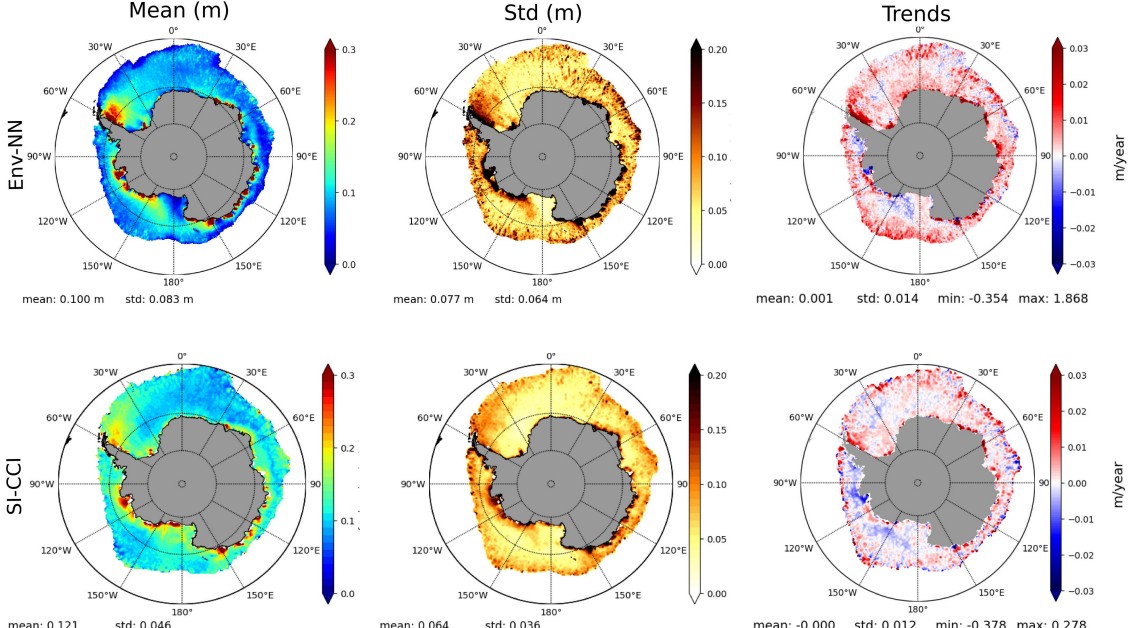

**Figure 4.** Comparisons between radar freeboard Envisat estimations computed from the neural network calibration (Env-NN, first line) and the SI-CCI Envisat solution (second line). The first column presents the 2003–2011 climatological mean maps and the second column the associated standard deviations. For the 2 datasets, maps of the 2003–2011 radar freeboard trends are provided in the third column.

### 3.2.2. Comparison against the SI-CCI Radar Freeboards

Figure 4 compares the climatological mean spatial distribution and variabilities of the calibrated Envisat radar freeboards with the SI-CCI dataset over 2003–2011. Maps of the trends are also provided. In addition, the 2011 monthly mean radar freeboard maps (from May to October), which present the seasonality of the spatial distributions are provided in Appendix A, Figure A1.

As for CS2, the SI-CCI radar freeboards tend to be thicker, with lower variabilities and comparable trends (although they are slightly lower). This calibration method well retrieves the pattern of thin new ice seawards the Ross ice shelf (cf. Section 3.1). Also, the nearly sea ice free zones of along coastal polynyas in the Weddell sea e.g., [77,78] are not represented in the SI-CCI data. More generally, the SI-CCI radar freeboard data are smooth, biased high, and rarely provide thin radar freeboards, compared to the Env-NN estimations. Over 2003–2011, the mean trend is positive, by about 1 cm per decade, which roughly represents 10% per decade (considering the 10 cm climatological mean radar freeboard). This mean tendency, as well as most of the spatial distributions, are comparable to that of CS2-TFMRA50, over 2011–2016 (cf. Figure 2). For CS2-TFMRA50, it represents about 16% per decade, with higher spatial differentiations. This could indicate that a slight intensification of the main regional trends, established since at least the beginning of Envisat observations, preceded the 2016 event. Note that the elongated pattern of positive trends along the coastal West Weddell sea is negative in the CS2 2011–2016 trend maps. It might be relevant to further investigate whether this reversal could have been a premise for subsequent changes in the Weddell sea. In the rest of the article, the acronym ENV will always refer to the ENV_NN calibrated estimation.

### 3.3. Comparisons with External Data

3.3.1. Sea Ice Mass Balance in the Antarctic (SIMBA) Transects

We analyse the sea ice freeboard and snow depth along transect lines conducted during the SIMBA campaign at 3 sites (Fabra, Brussels and Patria) of the Ice Station Belgica (ISB) drifting station, and at 3 short term stations (Stat1-3) in late September and October 2007. In-situ sea ice freeboards are calculated from the difference between the surface elevation, obtained by drilling auger holes, and the snow thickness, measured using a meter-long ruler. The snow and sea ice properties relative to the datasets are described in Lewis et al. [31].

For each station, Table 2 compares the mean in-situ transect measurements with the calibrated monthly mean Envisat sea ice freeboard data (ENV/AMSR) and the AMSR snow depth data. It also indicates the « extended SIT » which is computed from the sea ice freeboard and the snow depth mean data. In the case of negative ice freeboards, it is assumed that the « underwater » layer between the ice interface and the sea level is constituted of slush that has the property of the ice. Then, the total freeboard is the sum of the snow depth and a (negative) layer of slush under the seawater level (total freeboard = snow depth − ice freeboard). In addition, Figure A3, in Appendix A, specifies the comparison by indicating the results of each transect of the different sites.

**Table 2.** Mean sea ice freeboard and snow depth along each transect line of the SIMBA campaign. The ENV/AMSR column presents the Envisat sea ice freeboard that considers AMSR data to account for the radar speed decreasing into the snow. The SIMBA "Mean extended SIT" values are SIT estimations assuming that the layer between the ice interface and the sea elevation is only constituted of snow.

|  | Mean Ice Freeboard | | Mean Snow Depth | | Mean Extended SIT | |
| --- | --- | --- | --- | --- | --- | --- |
|  | ENV/AMSR | SIMBA | AMSR | SIMBA | ENV/AMSR | SIMBA |
| **Fabra** | 0.14 | −0.03 | 0.25 | 0.69 | 1.52 | 2.3 |
| **Brussels** | 0.12 | 0.05 | 0.27 | 0.07 | 1.41 | 0.55 |
| **Patria** | 0.12 | 0 | 0.25 | 0.33 | 1.44 | 0.9 |
| **Stat1** | 0.2 | −0.09 | 0.27 | 0.40 | 1.95 | 1.12 |
| **Stat2** | 0.2 | −0.09 | 0.28 | 0.52 | 2.02 | 1.45 |
| **Stat3** | 0.1 | −0.03 | 0.27 | 0.2 | 1.32 | 0.63 |
| **All stations** | 0.15 | −0.03 | 0.27 | 0.36 | 1.61 | 1.16 |

The Fabra site is mainly constituted of Multi Year Ice covered with thick snow layers, which results in flooding at the snow/ice interface. Therefore, the mean in-situ sea ice freeboard tends to be negative, while the snow thickness can reach up to 1 m. Envisat freeboard estimations are always positive (mean of 0.14 m, see Table 2), but are included within the large standard deviation intervals. This most likely indicates that the Ku-band radar penetrates, on average, ≈80% of the thick snow layer. The AMSR-E data underestimate the snow depth with a mean difference by about 45 cm, which represents a factor of 2.6. This is consistent with the 2.3 factor found by Worby et al. [33] in the East Antarctic sector. Our calculation gives a mean in-situ sea ice thickness of 2.30 m, which is consistent with the 2.34 m indicated in Lewis et al. [31]. This leads to altimetry sea ice thickness underestimation by about 80 cm (≈50%). Note that a mean in-situ ice thickness by ≈1.6 m is found if negative freeboards are set to zero, as it is sometimes done e.g., [79].

The Patria and Brussels sites are both considered as First Year Ice. The Brussels site has thin snow layers with mainly positive freeboards (no flooding), while Patria has much thicker snow which induces flooding events. At these 2 stations, the altimetry ice freeboards are comparable (with a mean of 0.12 m) and generally higher than the in-situ freeboards. At Brussels, where the snow is very thin, the Envisat mean freeboard is equal to the addition of the in-situ mean snow depth and freeboard (respectively equal to 0.07 and 0.05 m). This result is consistent with the study of Nandan et al. [80], that shows that Ku-band radar

echoes can not penetrate saline thin snow layers over FYI. At Patria, the altimetry data always overestimate the in-situ ice freeboard, indicating that the Ku-band radar should not completely penetrate the snow either. The mean altimetry ice freeboard is found at about 60% of the mean in situ snow depth. At Station 1, which is also constituted by FYI with a comparable snow thickness cover (leading to flooded snow-ice interfaces), the Envisat mean ice freeboard is exactly 50% of the mean in situ snow depth. These results fit with Willatt et al. [81] who found a mean dominant scattering surface of the Ku-band radar at about 50% of the snow depth.

Station 2 is considered as MYI covered with thick snow layer leading to surface flooding and therefore can be compared with the Fabra site. However, we found a freeboard overestimation. The signal seems to penetrate by about 60% of the snow layer, which is comparable to Fabra. This result may be linked with more flooding event and/or warmer surface temperatures.

The Station 3 is considered as FYI with medium thick snow layers associated with flooding. Mean sea ice freeboard is again overestimated from about 50% of the mean in situ snow depth.

Overall, we found that AMSR-E underestimates snow depth over thick and medium-thick snow layers, and strongly overestimates over thin layers. Very similar results are obtained when using the AMSR-clim dataset. The altimetry mean sea ice freeboards are always overestimated. It likely reflects both a radar echo mean scattering horizon within the snow layer, and the difficulty to retrieve freeboards in the presence of slush layer and flooding. By consequences, mean SIT altimetry estimations tend to be overestimated. However, it is important to recall that the SIMBA in-situ data account for a very local area, and at a particular time, with associated large standard deviations and uncertainties. In fact, the A-B sector is particularly affected by strong variabilities and small scale effects that can not be observed by the spatial and temporal resolution of altimetry measurements. Also, the warmer conditions of early spring probably affect the accuracy of measurements which increases the deviations with in-situ data.

### 3.3.2. ULS

This section compares the altimetry data with upward looking sonar (ULS) sea ice draft measurements in the Weddell sea [35]. This area is mainly composed of thick MYI with a large snow cover and important variability e.g., [82,83]. Table 3 presents the monthly means of ULS measurements and altimetric drafts computed using the TFMRA50 radar freeboards and the ASMR-E and AMSR-clim snow depths. All data from the period 2003–2011 are used to compute the climatological monthly means, standard deviations and RMSDs. For the comparison, the in-situ drafts are projected onto EASE2 grid.

Overall, the comparison shows relatively good consistency between the mean ULS data and the altimetric drafts. Using AMSR-clim, we found a mean overestimation by ≈11 cm (by ≈8%) from May to September and a mean underestimation by ≈40 cm (≈25%) in October. As they compensate each others, the annual mean deviation is only of a few centimeters. These results suggest that early spring modifications of snow and ice properties impact the satellite draft retrieval. This is coherent with the SIT underestimations detected at the Fabra site (which has comparable sea ice characteristics) during the in-situ September/October SIMBA campaign (cf. previous section). When using AMSR-E snow depths, we found a mean underestimation by ≈13 cm (≈9%) from May to September and by ≈55 cm (≈37%) in October. The annual mean deviation is of 20 cm (≈13%). This result tends to show that AMSR-E data underestimate snow depth over thick ice.

Table 3 also underlines large RMSD's, on average by about 1 m. In spite of coherent mean estimations, such deviations are the consequence of strong variabilities that can not be detected from monthly mean smoothed estimations. Compared to SIMBA, the greater consistency with ULS drafts is therefore most likely due to a larger and more representative statistical sample, that better fit with the space and time resolution of the satellite observations. Therefore, analyses presented in Section 3.4 rather consider large scale informations.

Note also that ULS data are known to potentially provide higher drafts because of the geometry of the underwater sea ice bottom.

**Table 3.** Comparison between climatological monthly mean ULS and Envisat drafts computed over 2003–2011. Envisat drafts are computed from the AMSR-E and AMSR-clim snow depths. For each month (05 to 10), we also indicate the climatological standard deviation (STD) and the Root Mean square Deviation (RMSD) with ULS drafts.

|  |  | 05 | 06 | 07 | 08 | 09 | 10 | **Mean** |  |
|---|---|---|---|---|---|---|---|---|---|
| ULS drafts | Mean | 1.35 | 1.33 | 1.47 | 1.66 | 1.61 | 1.52 | 1.5 | |
| | STD | 0.72 | 0.74 | 0.82 | 0.95 | 0.87 | 0.83 | 0.83 | |
| Envisat+AMSR-E drafts | Mean | 1.15 | 1.25 | 1.42 | 1.48 | 1.49 | 0.96 | 1.3 | |
| | STD | 0.91 | 0.8 | 0.87 | 0.94 | 0.84 | 0.83 | 0.87 | |
| | RMSD | 0.90 | 0.83 | 0.99 | 1.08 | 1.01 | 1.08 | 0.98 | |
| Envisat+AMSR-clim drafts | Mean | 1.40 | 1.47 | 1.64 | 1.76 | 1.72 | 1.13 | 1.52 | |
| | STD | 0.79 | 0.79 | 0.74 | 0.92 | 0.78 | 0.86 | 0.81 | |
| | RMSD | 0.96 | 0.8 | 0.94 | 1.06 | 0.97 | 0.96 | 0.95 | |

### 3.3.3. OIB

Table 4 and Figure 5 compare the mean total freeboards of Envisat, computed with AMSR-E (ENV-AMSRE) and AMSR-clim (ENV-AMSRclim), with the ATM airborne data for 6 OIB missions during the 2009 and 2010 campaigns (cf. Section 2.3). The 21 October 2009 and 30 October 2010 missions took place in the coastal Amundsen-Bellingshausen sector (CoA-B) and the 4 others in the Weddell sea. To analyse along-track variabilities, the satellite datasets are projected onto the OIB tracks, using a simple linear interpolation. Figure 6 presents the results for the 21 October 2009 and the 26 October 2010 missions. The comparison with the 4 other missions are provided in supplementary (Figure A4). The Envisat radar freeboards and the AMSR-E and AMSR-clim snow depths are also indicated in these figures. Note that a 12.5 km radius rolling mean smoothing is applied to OIB data in order to make comparisons at corresponding spatial scales [60].

**Table 4.** Comparison between mean OIB ATM total freeboards and Envisat drafts computed over 2003–2011. Envisat drafts are computed from the AMSR-E and AMSR-clim snow depths. (ab) indicate that the mission occurred in the coA-B sector and (w) in Weddel.

|  |  | 20091021 (ab) | 20091024 (w) | 20091030 (w) | 20101026 (w) | 20101028 (w) | 20101030 (ab) | **Mean** |  |
|---|---|---|---|---|---|---|---|---|---|
| ATM OIB | Mean | 0.32 | 0.42 | 0.45 | 0.46 | 0.49 | 0.69 | 0.47 | |
| | STD | 0.15 | 0.17 | 0.26 | 0.13 | 0.15 | 0.23 | 0.18 | |
| ENV+AMSR-E | Mean | 0.26 | 0.42 | 0.43 | 0.40 | 0.38 | 0.32 | 0.37 | |
| | STD | 0.16 | 0.22 | 0.27 | 0.19 | 0.15 | 0.15 | 0.19 | |
| | RMSD | 0.16 | 0.17 | 0.21 | 0.17 | 0.21 | 0.41 | 0.22 | |
| ENV+AMSR-clim) | Mean | 0.39 | 0.44 | 0.42 | 0.43 | 0.41 | 0.42 | 0.42 | |
| | STD | 0.13 | 0.17 | 0.21 | 0.15 | 0.14 | 0.16 | 0.16 | |
| | RMSD | 0.19 | 0.15 | 0.19 | 0.13 | 0.19 | 0.30 | 0.19 | |

Considering the 6 missions, these results indicate a mean freeboard underestimation by 10 cm for ENV-AMSRE and 5 cm for ENV-AMSRclim. In the CoA-B zone, OIB data show a strong inter-annual variability between the 2 years, with values in 2010 higher by about a factor 2. Mean total freeboards are consistent in 2009 but the increase in 2010 is not observed by Envisat. This behaviour might be related to what is observed in SIMBA transects (cf. Section 3.3.1). In the case of thick snow over flooded MYI, such as at the Fabra station, a strong snow depth underestimation leads also to a factor 2 higher in-situ observations. By contrast, where the snow is thin, the freeboard overestimations, likely due

to partial Ku-band penetration, compensate snow depth underestimations. It leads to more consistent Envisat mean freeboards. Comparisons with OIB radar freeboards would be necessary to better assess the impact of these changing conditions. In Weddell, the Envisat mean freeboards are comparable to OIB, with global slight underestimations of only a few centimeters ($\approx$3 cm using AMSR-clim and 5 cm using AMSR-E). Standard deviations of the datasets are comparable, which mostly reflects equivalent spatial scale resolutions (due to the rolling mean smoothing of OIB data). Still, the strong RMSDs logically illustrate small scale variabilities in OIB that are not captured by monthly mean satellite observations. The analysis of along-track variability (cf. Figures 6 and A4 in Appendix A) shows relatively good agreement (although Envisat freeboards can be biased, such as for the 30 October 2010 mission). As a result, considering the 6 OIB missions together, a correlation by about 0.5 is found (cf. scatterplots in Figure 5). It is coherent with validations in the Arctic.

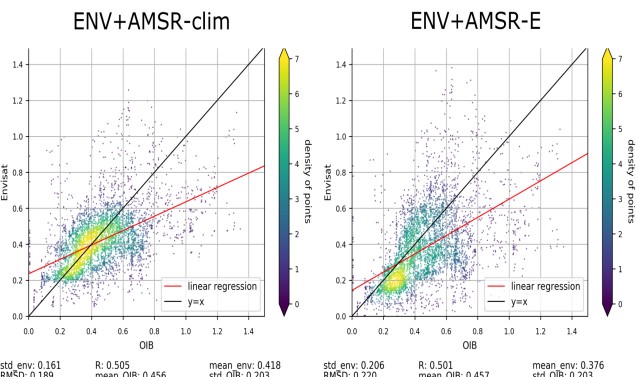

**Figure 5.** Scatterplots comparing the calibrated envisat total freeboards (using the 2 AMSR snow depth datasets) with all the data of the 6 OIB missions.

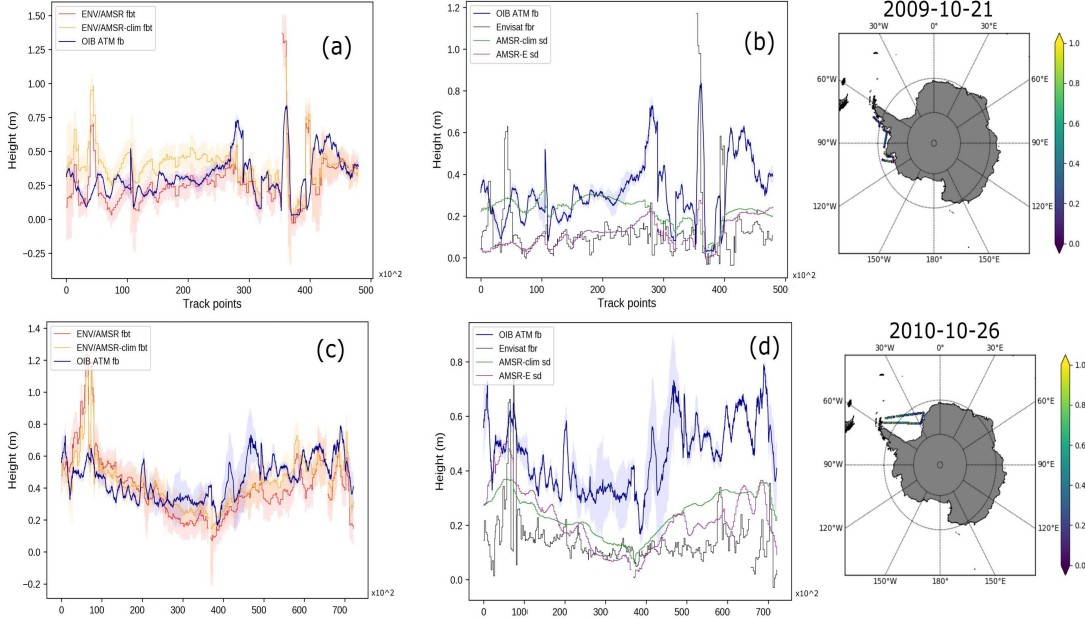

**Figure 6.** (**a**) Along-track comparison with the OIB ATM total freeboards of the 21 October 2009 (first line) and 30 October 2009 (second line) missions. The figures (**a**,**c**) compare with the calibrated Envisat total freeboards computed from AMSR-E (ENV-AMSRE) and AMSR-clim (ENV-AMSRclim). The figures (**b**,**d**) compare with the 2 AMSR snow depth solutions and the calibrated Envisat radar freeboards. The shaded zone refers to the uncertainty of the products. The maps on the third line indicate the geographical location of the OIB campaigns.

At the beginning of the 21 October 2009 mission, it is relevant to observe AMSR-clim snow depths comparable with ATM total freeboards. This might highlight the predominance of flooded ice, where zero ice freeboard should be observed. In this situation, the ENV-AMSRclim total freeboards are overestimated. This also tends to indicate a better retrieval using AMSRclim snow depths over thick snow layer. In the Weddell sea, this is very likely partly responsible for underestimations. AMSR-E snow depths are most of the time lower than AMSR-clim. It leads to bias low total freeboard estimations, which fits with results of Worby et al. [33].

### 3.3.4. ICESat-2

We compare total freeboard estimations of ICESat-2 with CryoSat-2 computed from CS2-TFMRA50 and ASD or AMSR-2 snow depth data. Figure 7 presents the annual mean maps of these two datasets for winter 2019.

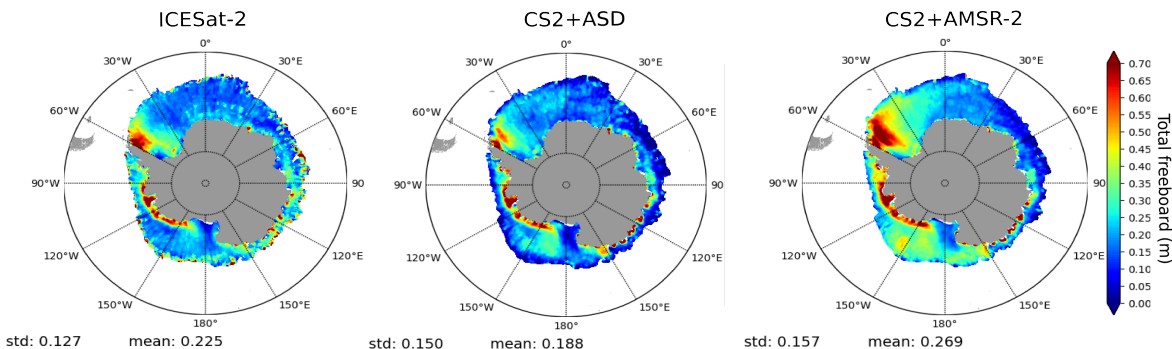

**Figure 7.** Maps of the annual mean total freeboards of ICESat-2, compared with CryoSat-2 estimations computed using ASD and AMSR-2 snow depth data in 2019.

Overall, the CS2 total freeboard estimations are in good agreement with ICESat-2 in terms of magnitude and spatial distributions. The use of AMSR-2 tends to overestimate the spatial mean by 4.5 cm, while the use of ASD leads to a mean underestimation by about 3.5 cm. In both cases, the correlation, computed from all the monthly data (instead from annual means), is about 0.55. Using AMSR-2, the mean overestimation is mainly the result of thicker freeboards in the Weddell sea and the A-B (and CoA-B) sectors. In particular, the pattern of thick freeboards drifting eastwards in Weddell is far too expanded compared to ICESat-2. On the contrary, the CS2+ASD freeboards are only very slightly underestimated and the extent is very similar. In fact, main deviations between ICESat-2 and CS2+ASD, responsible for the mean low bias, are at the ice/ocean transition areas. Everywhere, the total freeboard distributions and magnitudes are satisfactorily very similar. It highlights the potential of combining freeboard and snow depth from altimetry.

### 3.4. 2003–2020 Mean Trends

As already mentioned in the introduction, the sea ice extent data have shown a gradual slight increase until a record high in 2014. It was followed by an abrupt decline to reach the minimum of the past 40 years in 2017. In this section, we investigate the sea ice thickness and volume evolutions over 2003–2020. Figure 8, extracted from Kacimi and Kwok [54], specifies the regional sectors that are analysed. Table A1, in Appendix A, summarizes the trends for all sectors. Note that in order to provide a more comprehensive overview, trends are calculated considering data where the sea ice concentration (SIC) is greater than 75%, 50% and 15%.

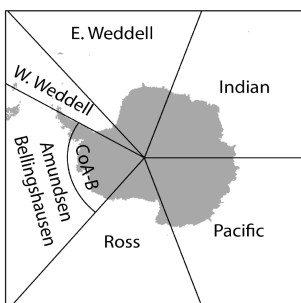

**Figure 8.** Geographical location of the different sectors. The figure is extracted from Kacimi and Kwok [54].

### 3.4.1. Global Analysis

Figure 9 presents annual mean time series of radar freeboard, snow depth, sea ice thickness and volume over 2003–2020 (for SIC > than 75%). The sea ice extent (SIE) and concentration (SIC) of the NSIDC product (cf. Section 2.3) are also indicated.

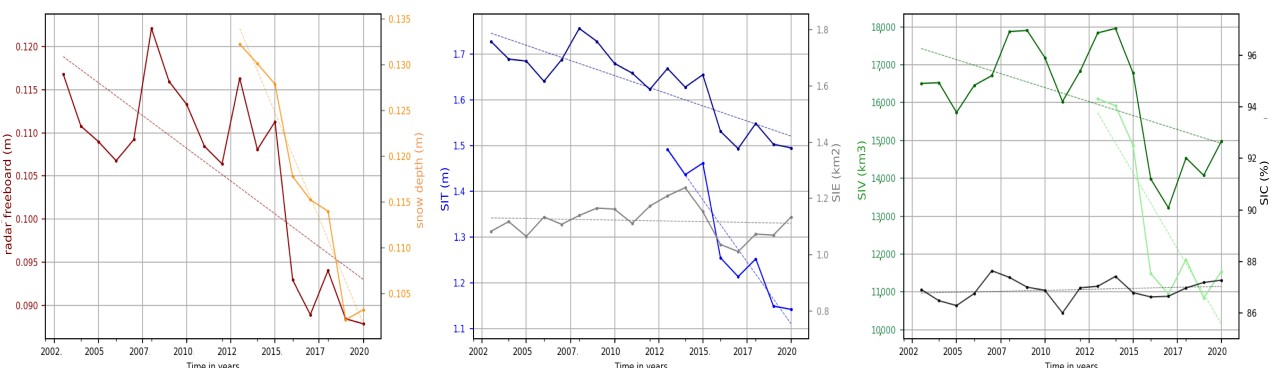

**Figure 9.** 2003–2020 time series of radar freeboard (in red), snow depth (in orange), sea ice thickness (in blue) and volume (in green) for sea ice concentration higher than 75%. The NSIDC sea ice extent (in grey) and concentrations (in black) are also represented. The radar freeboards are those of the CS2-TFMRA50 dataset. All 2013–2020 series are computed using the ASD snow depth while the 2003–2020 series use the AMSR-clim. Trends, with values indicated in Table A1, are also indicated in dotted lines.

Until 2015, we observe relatively stable mean radar freeboard values, by about 11–12 cm. The inter-annual SIE and freeboard variabilities do not reveal obvious first order correlations. On the contrary, the abrupt diminution of SIE, in 2016, is also clearly identified on the radar freeboard serie, with a coincident minimum in 2017. However, while sea ice cover has increased again after 2017, freeboards remained low, with the minimum of the serie reached in 2020. Overall, we find that mean sea ice radar freeboards have decreased by $\approx -14\%$ per decade over 2003–2020. This trend is much higher, about $-43\%$ per decade, over 2013–2020.

Regarding snow depth, the ASD data also show a gradual strong decrease, by about $-40\%$ per decade, since 2013. Since studies rather indicate recent positive precipitation anomalies (and future increasing of precipitation rates) e.g., [84–86], the snow thinning could be linked with changes in sea ice properties with, in particular, an increasing of snow-ice formation. Losses due to higher evaporation and sublimation related to higher air temperatures could also play a role.

As a result, we find that annual mean sea ice thickness and volume have strongly got thinner in the last decade. Since 2013, we detect a rate by $\approx -40\%$ per decade for SIT and $\approx -60\%$ per decade for SIV (using ASD). It is important to mention that these trends are mainly driven by the abrupt decline in 2016. Over 2003–2020, trends are smaller, by about $-10\%$ per decade for SIT and SIV (using the AMSR climatology). These long-term trends

are driven by radar freeboard and sea ice extent.Meanwhile, SIE and SIC trends are quite stable over 2003–2020 but the recent conditions lead to a reduction of SIE by ≈−17% per decade from 2013.

Despite that the sea ice extents in 2018–2020 are comparable to those of the « pre-2016 period », the mean radar freeboard, snow depth and then sea ice thickness and volume were kept low. This feature highlights the complex correlations between SIE and SIT, and suggests that the 2016 event could have initiated a long term decrease. In addition, winter SIT is also conditioned by summer conditions, that have rather decreased since 2013. Note that Figure A5 in Appendix A presents equivalent time series (as in Figure 9), but considering the smallest area with sea ice concentration higher than 50% over 2003–2020. Together with the trend values indicated in Table A1, it demonstrates that the results do not depend on assumptions about SIC.

Overall, from 2003 to 2015, the mean sea ice thickness and volume have globally increased. In 2016, the sea ice extent abrupt reduction is clearly associated with thinner sea ice, leading to a drastic sea ice volume reduction. Volumes have slightly increased thereafter, but values are still lower to the 2000's.

### 3.4.2. Regional Analysis

Figure 10 analyses regional time series of radar freeboard, snow depth, sea ice thickness and volume, over 2003–2020, in the Amundsen-Bell, the Coastal Amundsen Bell, and the Eastern and Western Weddell sea sectors (cf. Figure 8). Analyses of the other sectors are provided in Figure A6. As previously, the spatial annual means are computed for SIC > than 75% and Table A1 in Appendix A indicates the trends for SIC greater than 75%, 50% and 15%. This table confirms that the results in this section do not depend on assumption about SIC.

**Bell Amundsen:** The mean radar freeboard decreased by ≈−32% per decade over 2003–2020 and by ≈−65% per decade over 2013–2020. It is relevant to observe a maximum high in 2017, when the global mean sea ice extent is at its minimum. This maximum has preceded 3 years with the thinnest freeboards of the period. The minimum is reached in 2020. The snow depth variability follows well that of the radar freeboard. The ≈−47% per decade trend for 2013–2020 is mainly driven by the last three years of minima. As a consequence, the mean SIT has also decreased, by ≈−16% per decade over 2003–2020 and by ≈−43% per decade over 2013–2020. Current thicknesses are nearly twice lower than it was at the beginning of the envisat period. Overall, the volume has also decreased, by ≈−12% over the whole period. It has been nearly divided by two over 2013–2020. Meanwhile, the extent has also increased, by about 10% per decade over 2003–2020 but has decreased by nearly −60% since 2013. Note that the minimum in 2016 was not a particular event in this region.

**Coastal Bell Amundsen:** The mean radar freeboard has increased by ≈+22% per decade over 2003–2020 and by ≈+100% for 2013–2020. These trends are only driven by the period of 2013–2020 with no significant trends before. The maximum high freeboard was in 2011. It is followed by the minimum, in 2013. Since, the radar freeboards have increased, to reach maxima in recent years. Magnitudes are close to the maximum of the serie (in 2011). Unlike the previous "off-shore" Bell Amundsen zone, snow and freeboards are not well correlated, and snow depths do not show significant trends. The 2016 maximum low event is associated with a minimum of snow depth. Consequently, the mean SIT has also increased, by ≈+11% per decade over 2003–2020 and by ≈+66% per decade over 2013–2020. Logically, we do not observe significant trends for the extent (probably due to the absence of ice to ocean transition). Sea ice volumes have also increased, by ≈+10% per decade over 2003–2020 and by ≈+49% per decade over 2013–2020. An event with thick ice is also observed in 2011 but the maximum was reached in 2018. Recent thickness values are among the highest of the series. As in the Bell Amundsen sector, the 2016 event is not particularly perceptible.

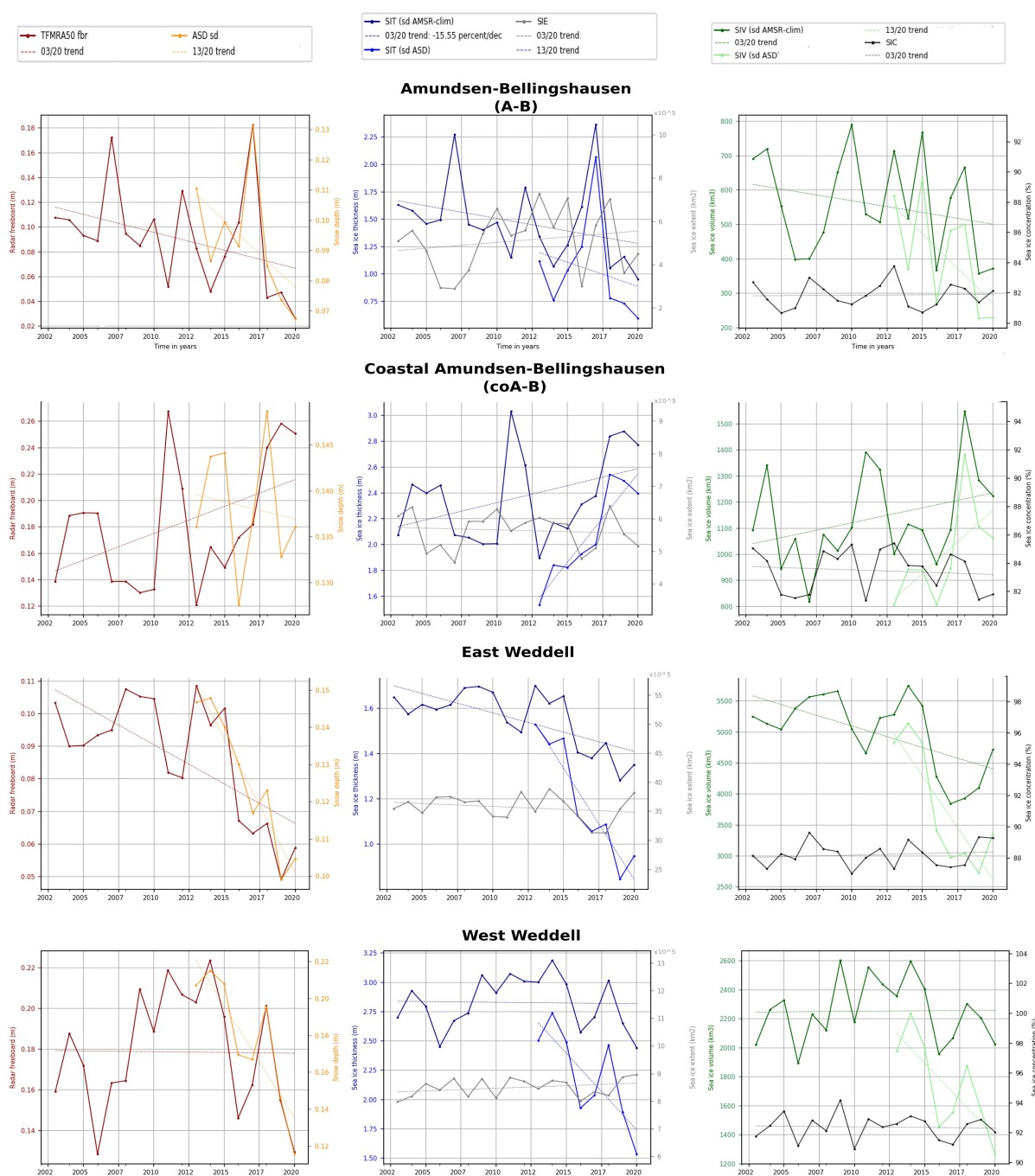

**Figure 10.** Same as for Figure 9 for 4 of the regions defined by Figure 8.

**East Weddell:** The radar freeboards decreased by ≈−28% per decade over 2003–2020 and by ≈−108% per decade from 2013. The "2016 event" is very well identified but does not represent the minimum since radar freeboard has continued to decrease thereafter. As a consequence the tendencies are particularly driven by the values after 2016. Trends over the period of Envisat are rather stable. Since 2016 snow depth is also strongly decreasing, by about −57% per decade. Then the sea ice thickness has decreased, by only ≈−10% per decade over the whole period, but by about −83% per decade since 2013. Before 2013, values where also quite stable. On the contrary sea ice extent and concentration have stayed relatively constant over the whole period, with highest values in recent years. However, the sea ice extent minima of 2016 and 2017 are well identified. As a consequence, sea ice volumes have strongly decreased since 2016. Our results indicates trends by nearly

−90% per decade since 2013. 2020 shows a slight increase but sea ice volumes are still at their lowest.

**West Weddell:** Considering the whole series, mean radar freeboards are relatively constant but they have decreased by ≈−55% per decade from 2013. However, the radar freeboard increases until 2016 and decreases thereafter. The lowest value, in 2020, is comparable to the minimum in 2006. Snow depths have also decreased, by ≈−68% per decade since 2013. The SIT was also stable over the whole period, with a decrease by about −60% per decade since 2013. These tendencies are associated with a slight positive trend of extent, with a well identified 2016 minimum. Overall, we do not find significant trends for volume. However, the decrease that was initiated by the high maximum in 2015, lead to a trend by ≈−55% per decade from 2013. This contrasts with the positive trends observed over the period of Envisat. Values are particularly low in the recent period.

## 4. Discussions

Interpreting the causalities for the various sea ice regimes, and their impacts on the uncertainties is challenging. Although it is now relatively well established that Ku-band cannot fully penetrates wet/moist/saline snow columns, the penetration depth remains a major limitation for altimetry-based freeboard estimations. This is particularly true in the Antarctic. Sea ice is generally warmer than in the Arctic and the absence of coastal boundaries surrounding the sea ice favours strong snow fall events (by the advection of northerly moist air masses). Especially, the resulting flooding events and the high salinity in the basal layers of the snow pack tend to reduce the Ku-band penetration. Another important source of error for freeboard is the sea ice roughness, that is not explicitly taken into account using the TFMRA. Deviations with SAMOSA+ probably provide first guess estimations of the impact of the roughness.

Overall, the data presented in this study generally overestimate the ice freeboard, and confirms the results suggested by Price et al. [45] and Kwok and Kacimi [53]. Kwok and Maksym [43] still argue that a zero ice regime may not be dominant and that this assumption, analysed in e.g., Kurtz and Markus [51], leads to important SIT underestimations. Although altimetry radar freeboards are likely biased, its knowledge remains essential for sea ice thickness. Additional studies, as for instance on the impact of snow temperature, density and/or surface roughness is yet necessary to characterize Ku band-radar penetration.

Our results have also highlighted that snow depth underestimations would counterbalance freeboard overestimations, leading to consistent total freeboards. The comparisons with ICESat-2 indicate that it is particularly the case when using ASD data. In fact, assuming a Ka-band reflection on the top of the snow pack, Ka-Ku dual frequency total freeboard estimations should be consistent, with deviations from validation data referring to the slush or the slush plus saline snow. The uncertainty on the SIT would cascade into the location of the ice/snow interface, which comes down to the difference in density between snow and ice. Of course it can only be the case with coincident Ka and Ku-band measurements. In this context, the upcoming CRISTAL mission, which will carry both Ka-band and Ku-band altimeters will surely provide more accurate total freeboard measurements.

Considering that ASD snow depth data better combine with Ku-band radar freeboards, a simple climatology, just like in the Arctic [60], could have been constructed. Unfortunately, the recent decrease of snow depth strongly limits the representativeness of such climatology in the past. This argues instead for the use of the AMSR-clim snow depth to construct time series including Envisat data.

Deviations with in-situ data (cf. Section 3) also showed that the complexity and high variability of the Antarctic sea ice occurs at space and time scales that can not be well captured by the current resolution of satellite altimeters. For this reason, the comparisons presented in this work rather consider large scale statistics (global to regional mean analysis).

The absence of ASD data prior to 2013 is an important obstacle to the analysis of long term sea ice volume trends and variabilities. The combination of AMSR-E/AMSR-2 data is not recommended as there is a bias between the 2 datasets. The 2003–2020 thickness trends (cf. Section 3.4) do not account for the snow depth inter-annual variations, which are therefore mainly driven by the radar freeboard. Considering that snow depth has decreased in the recent years, these trends are also very likely to be underestimated. The use of ASD to compute trends over 2013–2020 should improve their reliability.

Overall, our results clearly indicate recent sea ice freeboard and snow depth decreases. This reduction of snow depth might be the result of enhanced snow-ice formations, that results from sea ice thinning. In this case, the sea ice thickness data presented here are probably overestimated. Indeed, the Ku-band penetration depth would be reduced by an increase of slush layers and the rise of salt in basal layers resulting from snow-ice formation. This feature was well identified in comparisons with several SIMBA transects. As a consequence, the recent global negative trends of thickness could be stronger than indicated in this article. The snow depth decrease could also be the consequence of aerial redistribution blowing snow into leads, which reduces the snow that should have accumulated on sea ice. Although such phenomenon is more common than in the Arctic, it is very unlikely that it would reduce snow depth to the extent indicated in this study.

In the last part of this work, we have analysed global and regional time series by considering trends computed over 2013–2020. It demonstrates the recent sea ice behaviour, labelled by an important loss of sea ice volume. Of course, considering the very strong inter-annual variability of the Antarctic sea ice, these trends do not imply that winter sea ice will disappear in the next years. It is simply an effective manner to quantify the strong SIV loss that we found. Note that 2013–2020 is also the only period where we can take into account the inter-annual variability of snow depth. However, the significance of these trends, computed from 6 years of data, can be debated. As explain in this article, the source of uncertainties are so large that we have a relatively low confidence in the absolute magnitudes of thickness estimated from altimetry. Still, we can assume that the uncertainties do not drastically change from one year to another. Then, trends are much more relevant than magnitude to our analyse of recent SIT conditions.

## 5. Conclusions

In this paper, we presented the first altimetry-based sea ice freeboard, thickness and volume series over 2003–2020, at global and regional scales in the Antarctic. Such dataset is an important step towards an « Arctic-like » knowledge of the Antarctic sea ice, which is becoming an urgent need in the context of the recent warming events. The sea ice minimum extent record of the last summer 2021/2022 confirms this.

In the first part of this study, we presented sea ice radar freeboard estimates computed from CryoSat-2 heights measurements using a 50% threshold TFMRA retracker and the SAMOSA+ physical retracker. These 2 solutions have been compared with the SI-CCI dataset.

Overall, we found very consistent spatial distributions between the different datasets, with a 5–6 cm bias low compared to the SI-CCI product. In spite of equivalent global spatial means, the SAMOSA+ estimations have thinner thick freeboard patterns (like in the Weddell sea) and higher small freeboards. To determine the « best » radar freeboard solution remains challenging due to the absence of processed freeboard validation data. In our opinion, SAMOSA+ freeboards are probably more realistic but it can not yet be demonstrated in this study. In fact, ASD and AMSR snow depths are most often underestimated because: (1) the algorithm to retrieve snow depth from brightness temperatures has no sensitivity beyond about 50 cm of depth and (2) the Ku-band radar does not fully penetrate the snow, especially into wet, moist and saline snow layers such as in the Antarctic. Then, the TFMRA50 estimates are more appropriate to compute total freeboard (especially with the dual frequency ASD solution).

Envisat LRM measurements can not directly provide consistent freeboards from heuristic retracking, and physical retracker techniques are still under development. In order to produce consistent time series, the LRM Envisat radar freeboards are calibrated onto CS2 using a neural network approach trained over the common flight period. This methodology results in Envisat radar freeboards with a mean deviation from CS2 of the order of a centimeter. Nevertheless, this dataset has lower thick and higher thin freeboards, which is relevant with the estimation from SAMOSA+. Our analyses showed that such calibration is only efficient when using identical retracking techniques for CS2 and Envisat. Then, in this study, we have mainly focused on the TFMRA50 solution. Results from physical retracking are still very promising and should be considered in the future.

We analysed the spatial distributions of radar freeboard trends. It shows a global mean stability over the period of Envisat and a decrease over the period of CryoSat-2. In fact, relatively stable mean freeboards are observed until 2016. They decreased thereafter which is coherent with the abrupt sea ice extent decline in 2016. After 2016, the mean SIE has increased again to reach near average value in winter 2020. Meanwhile, the spatial mean freeboards remained low. Until 2016, the trends are characterised by rather positive patterns everywhere except in the Amundsen-Bell sector. After 2016, we observed a clear reversal of the trends. The most striking features are the reversal to positive trends in the A-B region and to negative trends in the Weddell sea. This suggests a change in sea ice regimes that could be the signature of the beginning of a sea ice melt period.

Freeboard and thickness estimates were compared with ULS draft data, some in-situ measurement of the SIMBA campaign and 6 missions of the 2009 and 2010 OIB campaigns. Overall, these comparisons mainly highlight mean radar freeboard overestimations and snow depth underestimations. These deviations are very likely related to the particular, and rapidly changing, sea ice conditions in the Antarctic (e.g., flooding or snow-ice formations). In such cases, the ku-band penetration is strongly reduced, leading to freeboard overestimations. However, such conditions are not always dominant. On the scale of the month, our comparisons show that regional means are generally quite consistent. At the SIMBA transects, the predominance of flooding event lead to higher freeboards (by about 15 cm). The AMSR-E data underestimate the snow depth by about 10 cm. The good statistical representativity of the ULS draft data sample lead to good global mean consistencies. Using AMSR-clim we found an identical mean draft and an underestimation by about 20 cm using AMSR-E. Comparisons with OIB also lead to consistent mean total freeboards. It also highlights that along-track variabilities of OIB can be quite well observed by Envisat. Still, specific events related to local temporary conditions can hardly be observed in monthly gridded estimations. It could be improved by using optimal interpolation techniques, that enables to consider time in the gridding process. Finally, the comparisons with ICESat-2 demonstrate very good consistency in terms of spatial distributions and magnitudes.

Our results indicate that the abrupt decline of sea ice extent in 2016 was associated with an important sea ice thickness and volume reduction. Since 2013, we found a diminution by about $-40\%$ per decade for thickness and by about $-60\%$ per decade for the volume. Unlike the sea ice extent, post-2016 thickness estimations remain low. Over the entire 2003–2020 period, the absence of non-climatological snow depth dataset does not allow for consistent trends. Note that they are still indicated in Table A1.

The regional analysis confirms a sea ice thickness and volume decreases everywhere except in the CoA-B sector, where the SIT has increased by about 65% (considering SIC > 50%). In all region except the CoA-B sector, 2016 is a thickness and volume maximum high. Except for 2016, we have not identified clear first order correlations between sea ice thickness and sea ice extent.

To go further, an analysis of correlations between sea ice summer conditions and the winter sea ice thickness would be of great interest. One other important point is to investigate the seasonal variability of sea ice. It should provide insights into whether 2016 impacted the timing of the melt and freeze. An update of the freshwater budget could also be made by using these estimations of sea ice volume variations. Finally, the datasets

presented and validated in this study provide a very important amount of information, aiming to impulse other studies in the Antarctic. It is crucial to better understand the climate alarm currently rising from the Antarctic.

**Author Contributions:** Conceptualization, methodology and software: F.G., M.B. and S.F., validation, F.R., G.G. and A.C., formal analysis, F.G.; investigation, F.G. and S.F.; resources, S.F.; data curation, S.F; writing—original draft preparation, F.G.; writing—review and editing, F.G., S.F., J.B., M.T., G.G., F.R. and A.C.; visualization, F.G.; supervision, project administration and funding acquisition: M.T., S.F. and J.B. All authors have read and agreed to the published version of the manuscript.

**Funding:** This work has received funding from The European Space Agency (ESA) in the context of a living Planet Fellowship and the Centre National d'Etudes Spatiales (CNES). CNRS contract number 969830.

**Data Availability Statement:** The sea ice and snow depth altimetric gridded monthly mean data are available at AVISO: https://www.aviso.altimetry.fr/en/data/products/altimetry-ice-products/altimetry-sea-ice-products-from-ctoh.html, accessed on 4 September 2022.

**Acknowledgments:** This research has received support by the CryoSat+ Antarctic Ocean projects (CSAO+), the CNES TOSCA CASSIS project and the ESA living planet fellowship. It has also benefited from the outcomes of the ESA POLAR+ Snow on Sea Ice and the Fundamental Data Record for Altimetry (FDR4Alt) projects. We also would like to thanks Stephan Kern, from AWI, for providing the AMSR climatology. We also thank the LEGOS laboratory and the reviewers for their relevant and constructive comments.

**Conflicts of Interest:** The authors declare no conflict of interest and the funders had no role in the design of the study; in the collection, analyses, or interpretation of data; in the writing of the manuscript; or in the decision to publish the results.

## Appendix A. Additional Figures and Tables

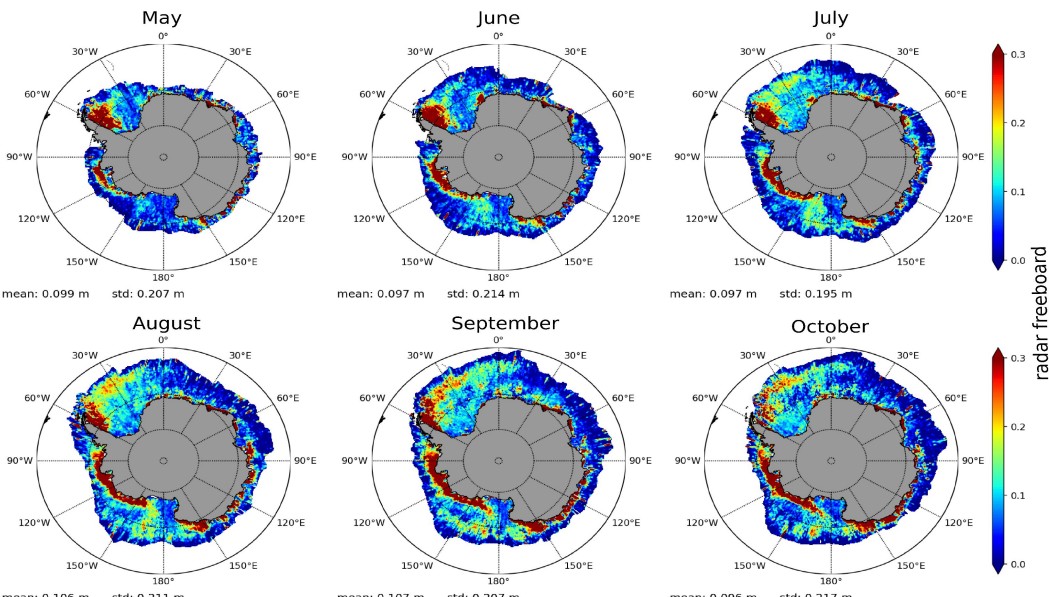

**Figure A1.** Monthly mean maps of CryoSat-2 radar freeboards in winter 2011. They are computed from the CS2-TRFMRA50.

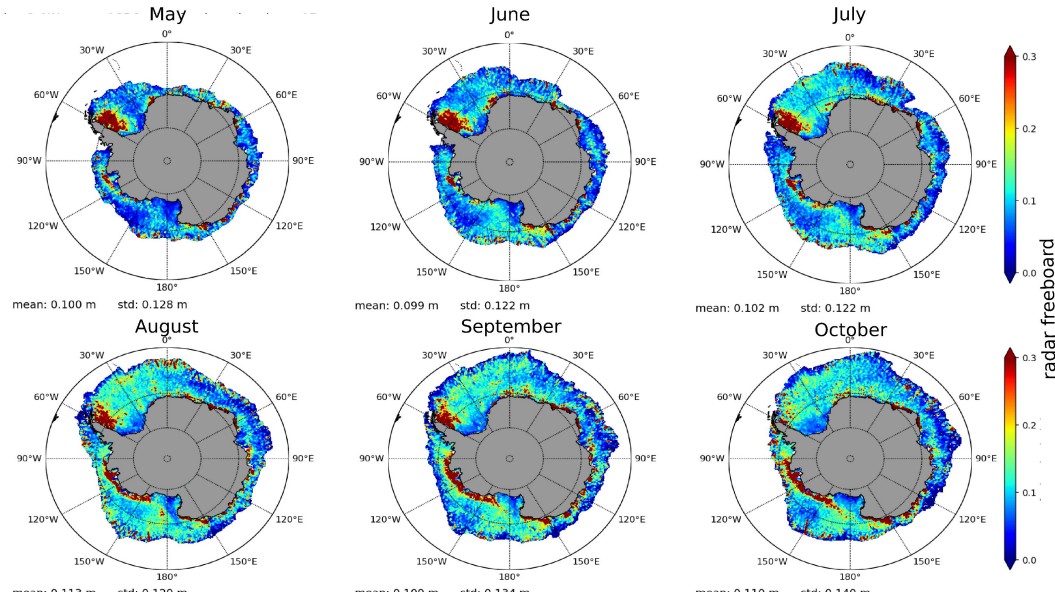

**Figure A2.** Monthly mean maps of the calibrated Envisat radar freeboards in winter 2011.

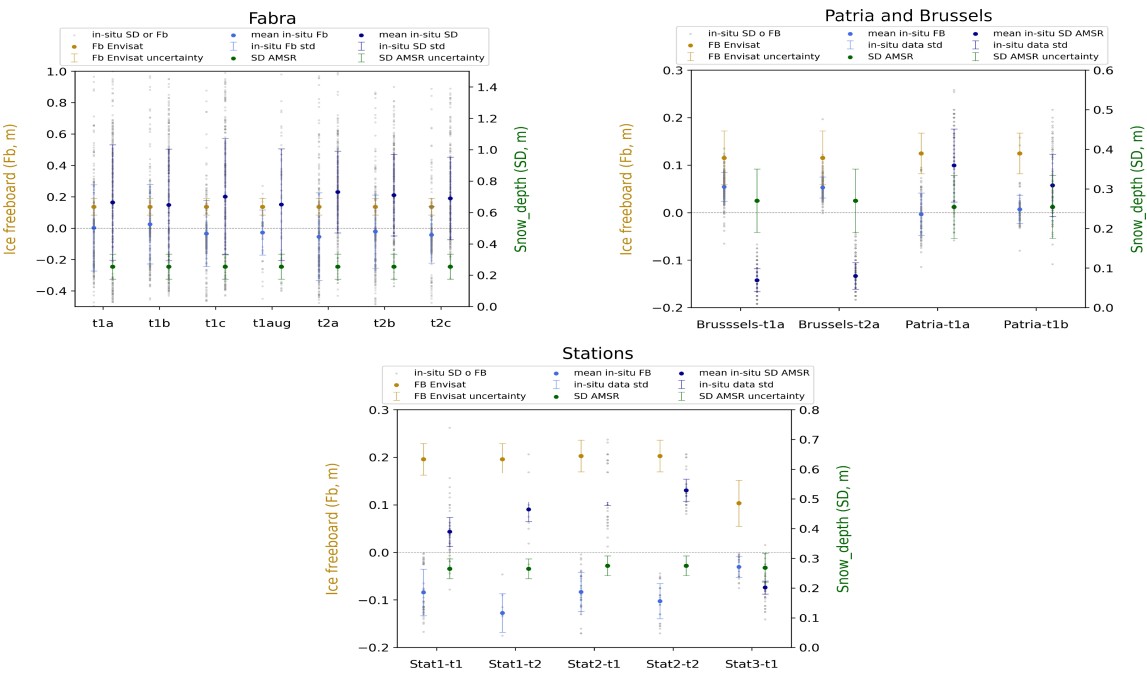

**Figure A3.** Comparisons between the sea ice freeboard and snow depth of SIMBA transects measurements with the Envisat sea ice freeboard and the AMSR snow depth data. For each transect the sea ice freeboards are presented on the first row, with values indicated on the left side Y-axis, and snow depths are presented on the second row, with values indicated on the right side Y-axis. The name of the transects are taken from Lewis et al. [31]. All the in-situ measures along the SIMBA transects are represented in grey with the mean value indicated in blue (clear blue for the ice freeboard and dark blue for the snow depth). the errorbars represent the ± standard deviation interval of the measurements. The range of uncertainties of the satellite data is also indicated by errorbars.

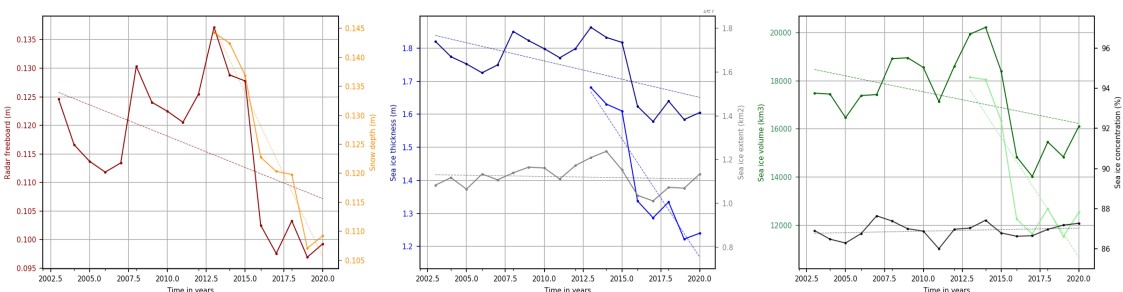

**Figure A4.** Same as Figure 6 for the 24 October 2009, the 30 October 2009, the 28 October 2009, and the 30 October 2010 missions.

**Figure A5.** Same as Figure 10 but with only the data of the smallest common area where sea ice concentration is always higher than 50% over 2003–2020.

**Table A1.** 2003–2020 and 2013–2020 trends (in % per decade) of the variables analysed in Section 3.4. The radar freeboards trends are calculated from the CS2-TFMRA50 dataset and the snow depth from the ASD product (only for 2013–2020). The SIT and SIV trends are computed from CS2-TFMRA50+AMSR-clim over 2003–2013 and from CS2-TFMRA50+ASD over 2013–2020. The NSIDC Sea ice extent and concentration trends are also indicated. The trends are calculated for sea ice concentration higher than 75%, 50% and 15%. Only for the global Antarctic, trends are also calculated for the smallest area with concentration higher than 50% over 2003–2020 (small 50%). It corresponds to the trends of the series presented in Figure A5.

| | | Fbr | | Sd | | SIT | | SIV | | SIE | | SIC | |
|---|---|---|---|---|---|---|---|---|---|---|---|---|---|
| | | 03-20 | 13-20 | 03-20 | 13-20 | 03-20 | 13-20 | 03-20 | 13-20 | 03-20 | 13-20 | 03-20 | 13-20 |
| **A-B** | 75% | −32 | −65 | x | −47 | −16 | −43 | −12 | −98 | +10 | −59 | ≃ | −1 |
| | 50% | −49 | −52 | x | −43 | −21 | −31 | −13 | −94 | +6 | −52 | ≃ | −2 |
| | 15% | −53 | −83 | x | −57 | −20 | −45 | −4 | −89 | +5 | −39 | ≃ | −8 |
| **CoA-B** | 75% | +22 | +103 | x | −3 | +11 | +66 | +10 | +49 | −2 | −12 | ≃ | −5 |
| | 50% | +37 | +113 | x | +5 | +17 | +71 | +15 | +66 | ≃ | ≃ | −1 | −5 |
| | 15% | +37 | +113 | x | +6 | +17 | +71 | +15 | +66 | ≃ | ≃ | −1 | −5 |
| **Ross** | 75% | −24 | −37 | x | −24 | −13 | −30 | −11 | −48 | ≃ | −18 | ≃ | −1 |
| | 50% | −27 | −46 | x | −25 | −13 | −33 | −13 | −49 | −2 | −16 | +1 | −1 |
| | 15% | −30 | −53 | x | −27 | −15 | −35 | −15 | −50 | −2 | −16 | +1 | −2 |
| **Pac** | 75% | +21 | −46 | x | −59 | +5 | −56 | +11 | −96 | +6 | −37 | ≃ | −2 |
| | 50% | +20 | −60 | x | −66 | +6 | −63 | +9 | −84 | +3 | −16 | +1 | −5 |
| | 15% | +18 | −60 | x | −64 | +6 | −60 | +8 | −80 | +2 | −4 | +1 | −5 |
| **Ind** | 75% | −48 | −23 | x | −18 | −26 | −21 | −34 | −53 | −8 | −30 | ≃ | ≃ |
| | 50% | −51 | −43 | x | −19 | −26 | −25 | −34 | −50 | −6 | −21 | ≃ | −2 |
| | 15% | −55 | −50 | x | −19 | −26 | −26 | −35 | −49 | −6 | −19 | ≃ | −2 |
| **E.Wed** | 75% | −28 | −108 | x | −57 | −11 | −83 | −14 | −88 | −3 | −5 | ≃ | +2 |
| | 50% | −26 | −110 | x | −58 | −10 | −83 | −13 | −89 | −4 | −8 | ≃ | +1 |
| | 15% | −30 | −115 | x | −59 | −11 | −84 | −14 | −87 | −4 | −5 | ≃ | −2 |
| **W.Wed** | 75% | ≃ | −55 | x | −68 | ≃ | −59 | ≃ | −59 | +2 | +4 | ≃ | −1 |
| | 50% | +2 | −53 | x | −63 | +1 | −57 | ≃ | −60 | +1 | −1 | ≃ | ≃ |
| | 15% | +1 | −53 | x | −61 | ≃ | −56 | −1 | −60 | +1 | −2 | ≃ | +1 |
| **Global** | 75% | −14 | −43 | x | −39 | −8 | −42 | −9 | −62 | −1 | −17 | ≃ | ≃ |
| | 50% | −12 | −41 | x | −39 | −7 | −41 | −9 | −58 | −2 | −15 | ≃ | −1 |
| | small 50% | −5 | −51 | x | −47 | −5 | −50 | −7 | −66 | −2 | −13 | ≃ | −2 |
| | 15% | −15 | −46 | x | −40 | −8 | -42 | −10 | −57 | −2 | −13 | ≃ | −2 |

## Appendix B. From Waveforms to Thickness

Sea ice thickness estimations from altimetry are based on the freeboard methodology [87]. The procedure used here is fully detailed in Guerreiro et al. [56]. The procedure is very comparable to the methodologies commonly used to derive sea ice thickness from altimetric radar echoes in the Arctic e.g., [47,71,88,89]. Indeed, as a first step, the objective is not to produce specific data, but to evaluate the capacity of current techniques to provide estimates in the Antarctic.

The first step is to identify the sea ice leads and floes from the Pulse-Peakiness (*PP*) criteria: $PP = \frac{max(WF)}{\sum_{i=0}^{N}(WF_i)}$. Echoes with a $PP > 0.3$ are considered as leads and echoes with a $PP < 0.1$ are considered as floes. Values between 0.1 and 0.3 are discarded. The altimeter ranges (or heights) of leads and floes are derived from the waveforms using 2 retrackers: (1) the Threshold First Maximum Retracker Algorithm (TFMRA, Helm et al. [90]), which is a basic retracking algorithm based on the maximum power of the waveform (with a 50% threshold) and, for CS2 only, (2) the Sar Altimetry MOde Studies and Application

over ocean (SAMOSA+) physical retracker, that aims to fit the waveforms using a surface back-scattering physical model set up from several parameters such as the roughness or the main back-scattering horizon [91]. The rest of the processing chain remains identical for the 2 retracking methods.

Heights are calculated taking into account the altitude of the satellite, the DTU15 Mean Sea Surface (MSS) and the relevant geophysical corrections, as described in [60]. Note that whether geophysical corrections can differ from one product to another, it has nearly no impact on the freeboard retrieval [92]. Radar freeboard $fbr$ is simply the difference between the heights of the sea ice floes $H_{floes}$ and the heights of sea ice leads $H_{leads}$ : $fbr = H_{floes} - H_{leads}$. Prior to this calculation, $H_{lead}$ are linearly interpolated under the floes within a 25 km window. In the absence of leads in this interval, observations are discarded. Based on the National Snow and Ice Data Center (NSIDC) sea ice concentration product (cf. Section 2), freeboards in areas of less than 75% of sea ice are removed.

The radar freeboard uncertainties $\delta_{fbr}$ are estimated from Equation (A1) assuming that the errors are unbiased, uncorrelated and follow the Gaussian propagation law. Uncertainties on the leads $\delta_{H_{leads}}$ are calculated along-track from the statistical dispersion (standard deviation) of heights within a 12.5 km radius. Since sea ice topography can significantly vary, we assume that the statistical dispersion of floes is equivalent to that of the leads, as expressed in Equation (A2).

$$\delta_{fbr} = \sqrt{\delta^2_{H_{leads}} + \delta^2_{H_{floes}}} \tag{A1}$$

$$with \quad \delta^2_{H_{floes}} = \frac{\sigma_{H_{leads}}}{N_{floes}} \quad and \quad \delta^2_{H_{leads}} = \frac{\sigma_{H_{leads}}}{N_{leads}}. \tag{A2}$$

The conversion to sea ice freeboard $fb$ (Equation (A3)) aims at taking into account for the slower Ku-band wave propagation into the snow layer [88,93,94].

$$fb = fbr + sd\left(\frac{c_v}{c_s} - 1\right) \tag{A3}$$

with

$$c_s = c_v(1 + 0.51\rho_s)^{-1.5} \tag{A4}$$

where $sd$ is the snow depth, $c_v$ is the speed of light in vacuum and $c_s$ is the speed of light in snow defined from the relationship of Ulaby et al. [95] expressed in Equation (A4). Considering the same assumptions as for the radar freeboard, the sea ice freeboard uncertainties are calculated from Equations (A5) and (A6).

$$\delta^2_{fb} = \delta^2_{fbr} + (\delta_{sd} \times C)^2 + (\delta_{\rho_s} \times sd \times B)^2 \tag{A5}$$

with

$$C = 1 - (1 + 0.51\rho_s)^{(-1.5)} \quad and \quad B = -1.5 \times 0.51 \times (1 + 0.51 \times \rho_s)^{-2.5} \tag{A6}$$

The sea ice freeboards are converted into SIT assuming the hydrostatic equilibrium between snow covered sea ice and the ocean (Equation (A7))

$$SIT = \frac{\rho_w fb + \rho_s sd}{\rho_w - \rho_i} \tag{A7}$$

Following Kurtz and Markus [51], we assume that the sea ice and snow densities are seasonally varying in the Antarctic. In October $\rho_i = 875$ kg/m$^3$ and $\rho_i = 900$ kg/m$^3$ from May to September [40], while $\rho_s = 320$ kg/m$^3$ for May, $\rho_s = 350$ kg/m$^3$ for June to September and $\rho_s = 340$ kg/m$^3$ for October [96]. The sea water density ($\rho_w$) is always set to 1024 kg/m$^3$ [97].

Finally, by combining the same previous assumptions with Equations (A1), (A2) and (A5), the estimation of sea ice thickness uncertainties is given by Equation (A8).

$$
\begin{aligned}
\delta^2_{SIT} = {} & (\frac{\rho_w}{\rho_w - \rho_i})^2 \times \delta^2_{fbr} \\
& + (\frac{1}{(\rho_w - \rho_i)^2}(\rho_s^2 + C^2\rho_w^2)) \times \delta^2_{sd} \\
& + (\frac{\rho_w fb + \rho_s sd}{(\rho_w - \rho_i)^2})^2 \times \delta^2_{\rho_i} \\
& + (\frac{sd}{\rho_w - \rho_i})^2(1 + \rho_w^2 B^2) \times \delta^2_{\rho_s} \\
& + (\frac{fb(\rho_w - \rho_i) - \rho_w fb - \rho_s sd}{(\rho_w - \rho_i)^2})^2 \times \delta^2_{\rho_w}
\end{aligned}
\tag{A8}
$$

where $\delta_{fb}$ is a statistical dispersion calculated with Equation (A1) and $\delta_{sd}$ is provided by the product. The uncertainties on the sea ice and sea water densities are set to be similar to those over First Year Ice in the Arctic: $\delta_{\rho_i} = 35.7 \text{ kg·m}^{-3}$ [98] and $\delta_{\rho_w} = 0.5 \text{ kg·m}^{-3}$ [97]. To be consistent with Kurtz and Markus [51], we use $\delta_{\rho_s} = 7 \text{ kg·m}^{-3}$ [99] for the uncertainty of the snow density.

Finally, sea ice volume is computed using the NSIDC sea ice concentration (SIC) and extent (SIE). At each grid point $(i, j)$, the sea ice volume (SIV) is given from Equation (A9).

$$
SIV = \sum_{i,j}(SIT_{i,j} \times A \times SIC_{i,j})
\tag{A9}
$$

where $A = 12.5 \times 12.5$ account for the EASE-2 grid pixel size area.

**Appendix C. Regional Trends**

**Ross:** The mean radar freeboard has constantly decreased over the whole period. Our results indicate a decrease by $\approx -24\%$ per decade over 2003–2020 and at $\approx -37\%$ per decade from 2013. Snow depth variability follows the radar freeboard with a decrease by about $-24\%$ per decade over 2013–2020. Radar freeboard and snow depth minima are reached in 2018. The last few years show a slight increase but the values remain among the lowest. Then, the sea ice thickness has also decreased, by $\approx -13\%$ per decade over the whole period and $\approx -30\%$ per decade over 2013–2020. These tendencies are not correlated with sea ice extent or concentrations, that have been relatively stable. Sea ice volume has also decreased by $\approx -11\%$ per decade over 2003–2020 and by $\approx -48\%$ per decade since 2013. Sea ice volumes maxima in 2008 and 2013 are associated with slightly higher sea ice extents. On the contrary the lowest 2018 does not seem to be linked with sea ice extent or concentration anomalies.

**Indian:** The mean radar freeboard has decreased by $\approx -48\%$ per decade over 2003–2020 and by $\approx -23\%$ per decade over 2003–2020. We observe a gap in 2011 which should indicates a bias due to thin ice overestimation of the re-calibration. Then, tendencies over 2003–2020 should be overestimated in this small coastal area. Still, we can observe that values after 2011 seem much lower (2011 is CS2 also) with a minimum in 2019. The 2016 event is very well perceptible in snow depth with a clear gap in values in 2016. Since, snow depths have remained at their lowest. Overall, snow depth has decreased by $\approx -18\%$ per decade over 2013–2020. Similar conclusions can be drawn for sea ice thickness and volume that respectively have $\approx -21\%$ and $\approx -53\%$ per decade decreasing over 2013–2020. The recent minima seem associated with low sea ice extent.

**Pacific:** The mean radar freeboard has increased by $\approx +21\%$ per decade over 2003–2020 but has decreased by about $-46\%$ per decade since 2013. Snow depths have also decreased, by $\approx -59\%$ per decade over 2013–2020. Consequently, although sea ice thickness have slightly increased, by about $+5\%$ over the whole period, it has decreased by $\approx -56\%$ per decade since 2013. Sea ice volumes exhibit a comparable behaviour but recent high extents

lead to values on the order of magnitude to those of the Envisat period. A clear maximum in 2013, associated with high sea ice extents is present in all series. Then the 2013–2020 trends are mainly driven by this event.

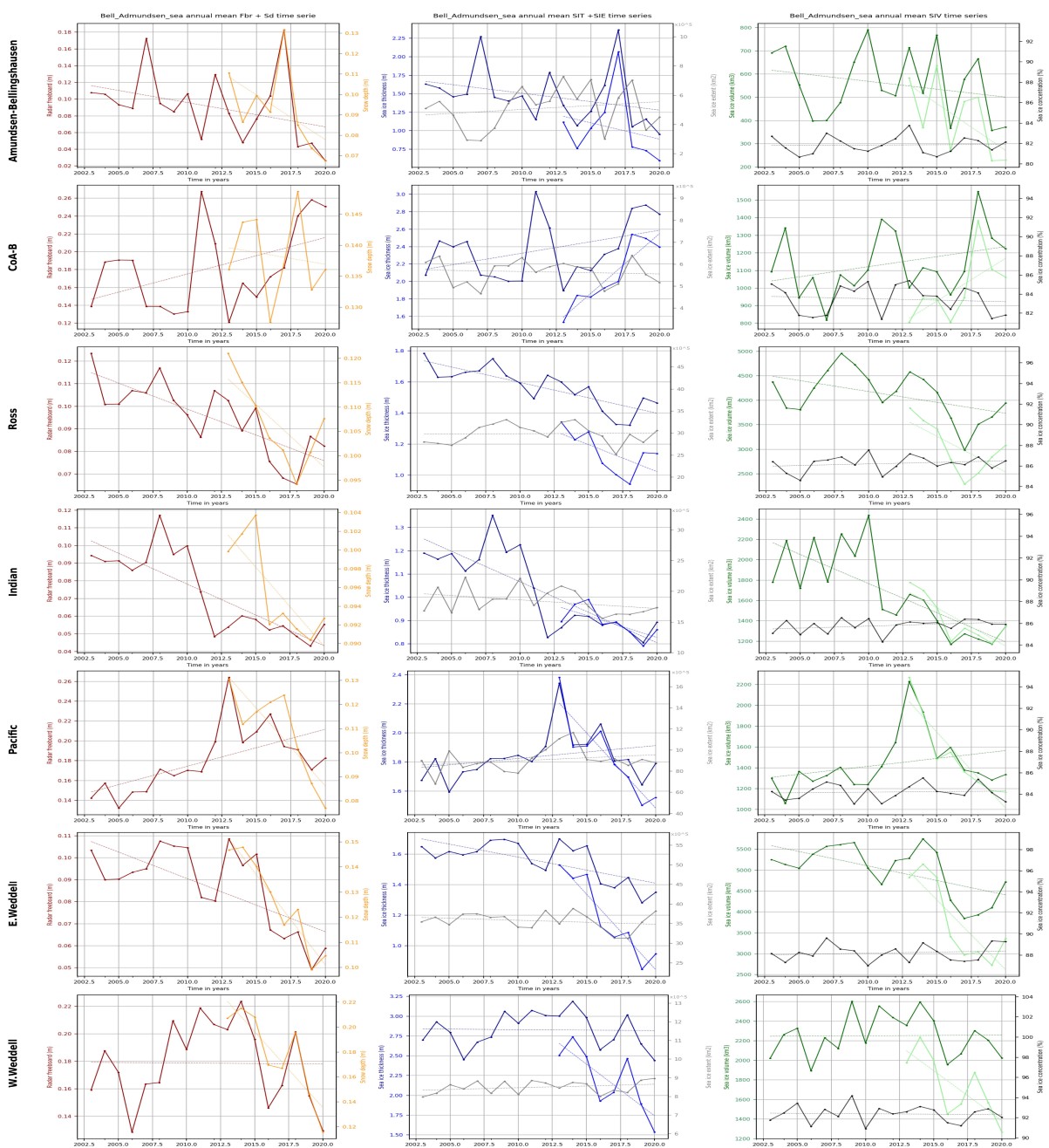

**Figure A6.** Same as for Figure 9 for the regions as defined in Figure 8.

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
