# Peer review of "Latest Altimetry-Based Sea Ice Freeboard and Volume Inter-Annual Variability in the Antarctic over 2003–2020"

_remotesensing, doi:10.3390/rs14194741_

Round 1

Reviewer 1 Report

Based on the sea ice observations from Envisat and CryoSat-2 during long time series, this manuscript uses neuronal networks to align these two datasets and recalculate the historical record of Antarctic sea ice freeboard, ice thickness and volume, and compared them with a variety of measured data, airborne altimeters and satellite altimeters, showing the high accuracy of the sea ice parameters obtained using the method of this manuscript. Further, historical Antarctic sea ice observations were analyzed to obtain trends in Antarctic sea ice change from 2003 to 2020. In general, the results of this manuscript are reliable, the overall framework is reasonable, innovative and well written. However, there are still some minor content and textual issues that require further revision.

You should explain the abbreviations of terms that appear for the first time in the abstract or in the text, e.g. in the abstract, Altimetric Snow Depthshould immediately follow ASD, etc. There are multiple other instances of this misrepresentation throughout the text.

Line 51 What is this sentence trying to say and it seems abrupt to put it in this position。

Line 101 There are references to the TFMRA and SAMOSA+ retrackers, which you should briefly describe in the introduction to the method.

Whether there is an overlap of orbits in the Antarctic region during the overlapping mission cycles of CryoSat-2 and Envisat, and how to consider the impact of sea ice drift on satellite observation data.

Figure 1 is hard to read. The diagram is not very clear, the colour bar on the right stands for "radar freeboard"? But the left-hand side is marked with "mean" and "sd" respectively. The first two columns seem to be statistics of CS-TFMRA-50 and CS-S+. How are the SD values calculated? The third column are comparisons of SI-CCI and CS2-TFMRA50, or CS2-S+? Should make it clear. And the values in the figure need explanation in the text.

147: IceSat-2 à ICESat-2

L189: may à May

Line 191 How to interpret the "climatological mean difference" and how it differs from the mean difference between the two data sets?

L213: the method to compute freeboard trends should be elaborated.

Figure 2. The rate of change of CS2 S+ in drywall for the two intervals 2011-2015 and 2016-2020 should be shown Figure 2. as well, in order to compare more clearly the differences between the two methods.

Section 4.1 The Envisat radar freeboard re-calibration algorithm is the highlight of this manuscript and although it is quoted from another manuscript, which has not yet been published, you should describe the steps of this algorithm in this manuscript in as much detail as possible.

Figure 4 The first column have no colour bars.

Line 284 Negative ice freeboard is treated here by simply considering slush as part of the sea ice and calculating the total drywall using total freeboard = snow depth - ice freeboard. However, the penetration of radar waves into the snow and the delay of radar waves in the snow lead to the introduction of a double uncertainty in the calculation of ice freeboard, which is a factor that must be considered when calculating ice freeboard using microwave (Ku Band) altimeter data.

Table 2. How is the "extended SIT" calculated here? The assumption of " the layer between the ice interface and the sea elevation is only constituted of slush that has the property of the ice " for negative ice freeboard in the preceding text is mentioned again here differently-“constituted of snow”. How can this be explained?

Three metrics are commonly used in the manuscript for comparisons between two datasets: Mean, STD and RMSD; usullly one of STD and RMSD is sufficient, as they can both represent differences from the validation data.

Figure 5. too many subgraphs in Figure 5. The caption is hard to follow. Subgraphs can be labeled with (a) (b) (c)….. Also for others figures.

Figure 7 What is the meaning of "2005.0" in the horizontal axis? The same problem occurs in Figure 9.

Section 8 is Conclusion, and is repeated with Section 9.

Line573: [?], correct it.

The Conclusion section should be improved,it’s too long. The current version looks like a brief review of the research work.

Reviewer 2 Report

In this manuscript the authors perform a new estimation of sea ice freeboard, thickness and volume from altimetry over the Envisat/CryoSat-2 2003-2020 time period and to identify first order impacts of the recent sea ice conditions. The topic is interesting and the manuscript is well written.

Reviewer 3 Report

The manuscript is dedicated to the altimetry-based sea ice freeboard and volume inter-annual variability in the Antarctic. This study is essential to provide sea ice thickness and volume estimations in order to anticipate potential multi-scale changes of the Antarctic sea ice, and its impacts on the entire climate system. The main objectives of this work are to assess a new estimation of sea ice freeboard, thickness and volume from altimetry over the Envisat/CryoSat-2 2003-2020 time period, and to identify first order impacts of the sea ice recent conditions. The results obtained are new, interesting, and valuable for the field. The results are clear, and their discussion is also presented well. Nevertheless, the paper is written with some disadvantages, and so it should be reorganized to be in accordance with the journal requirements to the structure of the article and its content. So, my opinion is that the paper needs major revision. I also suggest the authors make the following corrections before its publication:

1. The paper should have the following structure (see the journal template): Introduction, Materials and Methods, Results, Discussion, and Conclusions. Please, reorganize your paper in accordance with the required structure to provide the section Materials and Methods. It is also strange the article in the current form has two sections with the similar titles: ‘7. Discussions’ and ‘8. Discussion’.

2. Please, prepare the article exactly with the journal template. If not understandable, please, use any published paper as the example from: https://www.mdpi.com/journal/remotesensing.

3. Please, provide e-mail addresses for all the authors. Please, capitalize the first letter in the first author last name.

4. Section References is missing in the article. All the references in the article text looks like ‘?’, e.g., references in the first paragraph in Introduction looks as: ‘Observations of sea ice thickness (SIT) and concentration (SIC) in the Arctic have been extensively analysed in the last decades [e.g., ? ? ? ? ]. They have demonstrated the accelerating sea ice decline [e.g., ? ? ], its link with the global warming [e.g., ? ? ], and have emphasised the impacts on the whole climate system [e.g., ? ? ? ? ]. In the Antarctic, the situation contrasts with that in the Arctic. The ≈ 40 years of sea ice extent from passive microwave satellite measurements have recorded a global decades-long slight increase, to reach a record high in 2014 [? ? ]. This gradual positive trend, of ≈ 1.5% per decade, have large regional differences and variabilities. In particular in the Amundsen-Bellingshausen (A-B) sector, where the sea ice has rather reduced [? ]. This decrease, associated with an increase in the Ross Sea, is very likely to be linked with a deepening of the Amundsen Sea Low (ASL) pressure system that drives warm northerly flows over the A-B area and cold winds in the Ross Sea [e.g., ? ? ]. Although anomalies of meridional winds such as those associated with the El Nino-southern Oscillation (ENSO) and the Southern Annular Mode (SAM) are important factors to explain the sea ice interannual to multi-decadal variability [e.g., ? ? ], the global increase seems to have mainly been driven by the negative phase of the Interdecadal Pacific Oscillation [? ]. Explanations based on enhanced basal melting of ice shelves [? ] or the stratospheric ozone depletion [? ] have been denied by ? ] and ? ].’.

5. A pdf file of the article has no visible figures. All the figures looks like empty squares with figure file titles, e.g., Figure 1 looks like a square with the text ‘figs_ant/fig1_fbr.png’ in it. Please, provide the article with the visible figures.

6. The text in section ‘8. Discussion’ is the same as in section ‘9. Conclusions’. So, the text in Section ‘8. Discussion’ of the current version of the article should be removed completely.

7. Please, rewrite Author Contributions in accordance with the journal requirements:

Author Contributions: For research articles with several authors, a short paragraph specifying their individual contributions must be provided. The following statements should be used “Conceptualization, X.X. and Y.Y.; methodology, X.X.; software, X.X.; validation, X.X., Y.Y. and Z.Z.; formal analysis, X.X.; investigation, X.X.; resources, X.X.; data curation, X.X.; writing—original draft preparation, X.X.; writing—review and editing, X.X.; visualization, X.X.; supervision, X.X.; project administration, X.X.; funding acquisition, Y.Y. All authors have read and agreed to the published version of the manuscript.” Please turn to the CRediT taxonomy for the term explanation. Authorship must be limited to those who have contributed substantially to the work reported.”.

So, the paper needs major revision.

Reviewer 4 Report

Review on “New altimetry-based sea ice freeboard and volume inter-annual variability in the Antarctic over 2003-2020”, by Florent Garnier, Marion Bocquet, Sara Fleury, Jérôme Bouffard, Michel Tsamados, Frédérique Remy, Gilles Garric, and Aliette Chenal, submitted for publication in Remote Sensing.

General comments :

The paper studies the sea ice conditions in the Antarctic over the year 2003-2020 using remote sensing altimetry observations. The authors present different data processing techniques and their impact on the results. The paper is generally well written, but sections 7, 8, and 9 should be combined somehow because there are a lot of repetitions. Section 8 (second discussion) is quite similar if not identical to section 9 (conclusion). Maybe the authors did not submit their final version of the manuscript ? The paper is very long, with an extensive list of references. Overall, it is an interesting read for those working in or very familiar with the field of sea ice estimation from remote sensing data.

Specific comments :

Abstract, line 18: ASD has still not been defined at this point. Actually, many acronyms are used in the abstract but are not defined, except for LRM. Why define LRM but not the others ?

Line 170: “20007” should be “2007”.

Line 219: “northwester” should be “northwestern”.

Line 284: Shouldn’t it be (total freeboard = snow depth + ice freeboard) ? A figure would help here.

Line 301: “brussels” should be “Brussels”.

Line 311: “This results” should be either “This result” or “These results”.

Table 3: What are the columns from 05 to 10 ? Years ? The caption mentions the 2003-2011 time period. Please explain.

Line 353: “using simple a linear interpolation” should be “using a simple linear interpolation”.

Line 420: “SIT and SIV the (using the AMSR climatology)” should be “SIT and SIV the (using the AMSR climatology)”.

Line 422: SIE has not yet been defined at this point in the text.

Line 425: “depth and then sea ice thickness” should be “depth and then sea ice thickness”.

Line 433: “sea ice thickness and volume has globally increased” should be “sea ice thickness and volume have globally increased”.

Line 440: “SIC > than 75%” should be either “SIC > 75%” or “SIC greater than 75%”.

Line 453: “On the meanwhile, the sea ice extent have also increase, of about 10% per decade over 2003-2020 but have decreased…” should be “Meanwhile, the sea ice extent has also increase, by about 10% per decade over 2003-2020 but has decreased…”. Generally, in the text, change “of x%” to “by x%”.

Lines 468, 481 and 482: “serie” should be “series”. Also the line 614.

Line 484: “Sea ice thickness is was also stable” should be either “Sea ice thickness is also stable” or “Sea ice thickness was also stable”.

Line 488: “The recent period is also marked but particularly low values” should be “The recent period is also marked by particularly low values”.

Line 535: “thickness trends could be even be stronger than…” should be “thickness trends could be even stronger than…” or “thickness trends could even be stronger than…”

There are 2 discussion sections (section 7 and section 8)

Line 573: Fill in the question mark.

Line 603: The comparisons with ICESat-1 has not been presented in this paper. If so, where ?

Figures A1 and A2: please specify the month for each map.

Figure A4: Please use consistent date format in the caption.

Round 2

Reviewer 3 Report

Dear authors,

Thank you very much for your manuscript revision. Its current version looks much better than the previous one, and can be published as is.

So, the paper can be accepted in present form.

Author Response

Dear Reviewer 3.

Thank you very much for your review. We are very pleased to hear that the revised version of the manuscript meets your expectations.

Kind regards